# The case of a southern European glacier which survived Roman and Medieval warm periods but is disappearing under recent warming

Ana Moreno[1], Miguel Bartolomé[2], Juan Ignacio López-Moreno[1], Jorge Pey[1,3], Juan Pablo Corella[4], Jordi García-Orellana[5,6], Carlos Sancho[Ŧ], María Leunda[7,8], Graciela Gil-Romera[9,1], Penélope González-Sampériz[1], Carlos Pérez-Mejías[10], Francisco Navarro[11], Jaime Otero-García[11], Javier Lapazaran[11], Esteban Alonso-González[1], Cristina Cid[12], Jerónimo López-Martínez[13], Belén Oliva-Urcia[13], Sérgio Henrique Faria[14,15], María José Sierra[4], Rocío Millán[4], Xavier Querol[16], Andrés Alastuey[16] and José M. García-Ruíz[1]

1. Departamento de Procesos Geoambientales y Cambio Global, Instituto Pirenaico de Ecología – CSIC, Zaragoza, Spain
2. Departamento de Geología, Museo de Ciencias Naturales - CSIC, Madrid, Spain
3. Fundación Aragonesa para la Investigación y el Desarrollo, ARAID, Zaragoza, Spain
4. CIEMAT — Environmental Department (DMA), Avenida Complutense 40, Madrid, Spain
5. Institut de Ciència i Tecnologia Ambientals, Universitat Autònoma de Barcelona, Barcelona, Spain
6. Departament de Física, Universitat Autònoma de Barcelona, Barcelona, Spain
7. Institute of Plant Sciences & Oeschger Centre for Climate Change Research, Bern, Switzerland
8. Swiss Federal Research Institute for Forest, Snow and Landscape Research WSL, Birmensdorf, Switzerland
9. Department of Ecology, Faculty of Biology, Philipps-Marburg University, Marburg, Germany
10. Institute of Global Environmental Change, Xi'an Jiaotong University, Xi'an, China
11. Departamento de Matemática Aplicada a las TIC, ETSI de Telecomunicación, Universidad Politécnica de Madrid, Madrid, Spain
12. Centro de Astrobiología – CSIC-INTA, Madrid, Spain
13. Departamento de Geología y Geoquímica, Facultad de Ciencias, Universidad Autónoma de Madrid, Madrid, Spain
14. Basque Centre for Climate Change (BC3), Leioa, Spain
15. IKERBASQUE, Basque Foundation for Science, Bilbao, Spain
16. Institute of Environmental Assessment and Water Research – CSIC, Barcelona, Spain

Ŧ Deceased

**Corresponding author:** Ana Moreno (amoreno@ipe.csic.es) ORCID: 0000-0001-7357-584X

**Keywords**

Pyrenees, mountain glacier, current global warming, Medieval Climate Anomaly, Monte Perdido

**Abstract**

Mountain glaciers have generally experienced an accelerated retreat over the last three decades as a rapid response to current global warming. However, the response to previous warm periods in the Holocene is not well-described for glaciers of the southern Europe mountain ranges, such as the Pyrenees. The situation during the Medieval Climate Anomaly (900-1300 CE) is particularly relevant since it is not certain whether the southern European glaciers just experienced significant ice loss or whether they actually disappeared. We present here the first chronological study of a glacier located in the Central Pyrenees (NE Spain), the Monte Perdido Glacier (MPG), carried out by different radiochronological techniques and a comparison with geochemical proxies from neighboring paleoclimate records. The chronological model evidences that the glacier persisted during the Roman Period and the Medieval Climate Anomaly. The apparent absence of ice from the past ~600 years suggests that any ice accumulated during the Little Ice Age has since ablated. This interpretation is supported by measured concentrations of anthropogenic metals, including Zn, Se, Cd, Hg and Pb, which have concentrations well below those typical of industrial-age ice measured at other glaciers in Europe. This study strengthens the general understanding that warming of the past few decades has been exceptional for the past two millennia.

## 1. Introduction

Mountain glaciers are sensitive to climate variations on temporal scales from decades to centuries. It is well known that summer temperature and winter precipitation are the most important climate parameters influencing glacier mass balance (Oerlemans, 2001). Therefore, continuous records of past glacier size fluctuations provide valuable information about the timing and magnitude of Holocene climate shifts, which contributed to explain the characteristics and evolution of plant cover, human movements and land use (Solomina et al., 2015, 2016). Several glacier advances during the Neoglacial (which started around 6000-5000 years ago) have been identified (Bohleber et al., 2020) and associated to sustained cooling periods across the North Atlantic (Wanner et al., 2011). The most recent period of global glacier expansion took place during the Little Ice Age (LIA), beginning in the 13[th] century and reaching a maximum between the 17[th] and 19[th] centuries (Solomina et al., 2016). Afterwards, most glaciers worldwide have retreated rapidly, as indicated by measurements of changes in ice volume and ice-covered area, and this trend seems to have accelerated over the last three decades (Marzeion et al., 2014; Zemp et al., 2015, 2019).

Despite broad agreement on millennial-scale trends in global glacier fluctuations and Holocene climate variability (Davis et al., 2009; Solomina et al., 2015), regional variations are not so well constrained. The Pyrenees are a mountain range that currently hosts the majority of the southernmost glaciers in Europe. In this mountain chain there is a significant lack of knowledge about Holocene glacier fluctuations, with little evidence of Neoglacial advances (García-Ruiz et al., 2020). Based on Pyrenean tree-ring chronologies, summer temperatures during the Medieval Climate Anomaly (MCA, circa 900–1300 CE) were estimated to have been as warm as those of the 20[th] century (Büntgen et al., 2017), but no information is available on the glacier response to MCA warming. Conversely, glacier advance during the LIA is well constrained in the Pyrenees (García-Ruiz et al., 2014; González Trueba et al., 2008; Hughes, 2018; Oliva et al., 2018) and significant deglaciation is also evident in recent times (López-Moreno et al., 2016; Rico et al., 2017). In particular, the period from the 1980s to present has been the most intense in terms of the number of glaciers that disappeared (from 39 inventoried Pyrenean glaciers in 1984 to 19 at present; Rico et al. 2017). Given the

small size of the Pyrenean glaciers and their current critical situation in the context of global warming, we hypothesize that they could have disappeared completely during warm periods such as the MCA.

This study is focused on Monte Perdido Glacier (MPG), located in the Marboré Cirque in the Spanish Central Pyrenees. MPG is currently one of the most intensely monitored small glaciers (<0.5 km$^2$) worldwide (López-Moreno et al., 2016, 2019). Previous research based on different ground-based remote sensing techniques has demonstrated a rapid retreat of this glacier, with an average loss of ice thickness of about one meter per year since 1981 (López-Moreno et al., 2016, 2019). This glacier is located in one of the few valleys in the Pyrenees where information about Holocene glacier fluctuations exists. The outermost moraine in Marboré Cirque was recently dated at 6900 ± 800 $^{36}$Cl yr BP (García-Ruiz et al., 2020), which is the oldest Holocene date available for glacial deposits in Spain, and indicates a glacier advance during the Neoglacial period. Other minor advances would have occurred in MPG prior to the LIA, as inferred from three polished surfaces dated at 3500 ± 400, 2500 ± 300 and 1100 ± 100 $^{36}$Cl yr BP (García-Ruiz et al., 2020). Unfortunately, no information has been obtained on the glacier response to Roman or MCA warming periods, remaining an open question whether MPG just experienced significant ice loss or melted away totally. Most likely, the voluminous moraine at the foot of the Monte Perdido Massif was deposited during the LIA, indicating an important glacier advance. These results, together with evidence of long-term retreat from its LIA position indicated by pictures and moraines, suggest that this glacier could disappear over the next few decades (López-Moreno et al., 2016).

The present study aims to reconstruct the chronology of MPG ice sequence by using a variety of dating techniques and the analysis of several proxies associated with environmental and anthropogenic changes measured on a set of samples taken from a transect. Such analyses will fill the existing knowledge gaps and address the key question of whether Pyrenean glaciers may have survived previous Holocene warm periods.

**2. Study area**

The MPG (42°40′50″N; 0°02′15″E) is located in the Central Spanish Pyrenees, in the Ordesa and Monte Perdido National Park (OMPNP) (Fig. 1). It currently consists of two separate ice bodies, which were connected in the past. Both are north facing, lie on structural flats beneath the main summit of the Monte Perdido Peak (3355 m a.s.l.) and are surrounded by nearly vertical cliffs of 500–800 m in height under conditions of mountain permafrost (Serrano et al., 2020). At the base of the cliffs, the Cinca River flows directly from the glacier and the surrounding slopes, and has created a longitudinal west–east basin called the Marboré Cirque (5.8 km$^2$). This is the area within the Pyrenees with the highest variety of recent morainic deposits (García-Ruiz et al., 2014, 2020). Additionally, a 6 m thick sediment core obtained in 2011 from a lake inside the cirque (Marboré Lake) provided valuable information from the last 14,600 years of the depositional evolution of the lake (Oliva-Urcia et al., 2018) and of the regional variations in vegetation cover (Leunda et al., 2017). The Marboré Lake (2595 m a.s.l.) is located in the Marboré or Tucarroya Cirque, at the foot of the Monte Perdido massif. The distance between the lake and the MPG is approximately 1300 m and, therefore, both have been affected by similar past climate changes.

The total surface area of MPG in 2016 was 0.385 km$^2$, with an average decrease in glacier ice thickness of 6.1 m over the period 2011 - 2017 (López-Moreno et al., 2019). According to recent measurements of air temperature (July 2014 to October 2017), the 0 °C isotherm lies at 2945 m a.s.l., suggesting that the potential glacier accumulation area is very small, and perhaps non-existent, during warm years. The average summer (June to September) temperature at the foot of the glacier from 2014 to 2017 was of 7.3 °C (López-Moreno et al., 2019). No direct observations of precipitation are available from the glacier, but the maximum accumulated snow by late April in the three available years (2014, 2015 and 2017, when no scanning limitations occurred when the whole glacier was scanned) was 3.23 m, and field-measured average snow density was 454 kg m$^{-3}$, indicating that the total water equivalent during the main accumulation period (October to April) has recently been about 1.5 m (López-Moreno et al., 2019).

**3. Material and methods**

*3.1. Ice sampling and storage*

Ice sampling on MPG was carried out in September 2017 along a chrono-stratigraphical sequence from the lowermost and assumedly oldest to the uppermost and assumedly youngest ice preserved in the glacier, following the isochronal layers that emerge in the ablation zone (Fig. 2A). Vertical cores were not recovered because the glacier does not meet the usual glacio-meteorological and topographical criteria required to obtain a preserved ice-core stratigraphy. The unfulfilled criteria include low temperatures to prevent water percolation or a large extension and flat surface topography to minimize the influence of glacier flow (Garzonio et al., 2018). Samples were collected in an area with no evidence of major current ice movement, as confirmed by results from interferometric radar and GNSS measurements (López-Moreno et al., 2019). Due to the small size of this glacier and the absence of ice movement, we expected the ice to be frozen to the permafrost bedrock, and hence nearly stagnant, thereby reaching a substantial age as indicated by previous studies in similar glaciers (Gabrielli et al., 2016; Haeberli et al., 2004). The sampling sector lies in the ablation zone of the present-day MPG and has been eroded to form a current steady slope of 20° where it is possible to observe the primary stratigraphy, marked by clear debris-rich layers. The distribution of these debris-rich layers is rather regular and extends laterally (Fig. 2B), as would be expected for the primary stratification resulting from the original surface deposition of snow and debris. Therefore, these layers are considered isochrones, and confirm and facilitate the sampling along the slope, from the oldest to the youngest ice preserved in the glacier.

We measured one-meter thickness using a Jacob's staff at each sampling point along the slope (inset in Fig. 2A). The tilt of the ice layers was unclear but, since previous studies calculated about 30 m of ice thickness (López-Moreno et al., 2019), the ice layers probably dip steeply, as illustrated in Fig. 2A. After removing ~0.5 m of (possibly contaminated) surface ice, three or four horizontal cores, each of diameter 6 cm and length 25 cm, were sampled using a custom stainless-steel crown adaptor on a cordless power drill (Fig. 2C). Following this sampling procedure we recovered a total of 100 samples. The ice samples were stored in a freezer room at the IPE-CSIC in Zaragoza until they were melted and analysed to obtain their chronology combining

$^{210}$Pb, $^{137}$Cs and $^{14}$C techniques, and their geochemical composition in trace metals,
such as Pb or Hg (see below).
*3.2. Dating by using $^{210}$Pb and $^{137}$Cs.*
The isotope $^{137}$Cs, associated to the fallout from nuclear tests during the 1950s and the
1960s, as well as the Chernobyl (1986) (Haeberli et al., 1988) and Fukushima (2011)
nuclear accidents, was investigated by γ-spectrometry in five samples recovered
towards the top of the MPG chronological sequence (MP-61, MP-82, MP-97, MP-98,
MP-100, Table 1). In addition, ten samples were selected to perform a $^{210}$Pb analysis as
an independent dating method to obtain the age model of approximately the last
hundred years of glacier ice (Eichler et al., 2000; Herren et al., 2013). These samples
were selected also from the top of the ice sequence to collect the younger ice (Table
2). Determination of $^{210}$Pb activities was accomplished through the measurement of its
daughter nuclide, $^{210}$Po, by α-spectrometry following the methodology described in
(Sanchez-Cabeza et al., 1998) (Table 2).
*3.3. Dating by $^{14}$C method.*
Sixteen accelerator mass spectrometry (AMS) $^{14}$C dates from MPG ice were obtained
by combining bulk organic matter (9 samples), pollen concentrates (3 samples), bulk
sediment accumulated in filters (2 filters), and water-insoluble organic carbon (WIOC)
particles (2 samples) (Table 3). The procedure to select these samples was as follows:
(i) Using a binocular microscope [x10], we picked up organic particles for dating from
selected ice samples. However, the small size of the handpicked organic remains
prevented us from classifying them. As a result, we obtained 9 samples (MP-1, MP-42,
MP-48, MP-67, MP-68, MP-69, MP-70, MP-73, MP-100, Table 3) that were sent to the
Direct AMS laboratory (Seattle, USA) for dating. The selection of those nine samples
was based on the amount of debris found in the sample, once the ice was melted.
(ii) Pollen concentrates were prepared from three samples (MP-30, MP-70 and MP-
100) to complete the previous set with the aim of replicating some of the results (MP-
70 and MP-100) and obtaining new dates (MP-30). Preparation followed the standard
palynological method, including a chemical treatment and mineral separation in heavy

liquid (Thoulet: density 2.0; Moore et al., 1991). The effects of meltwater percolation on pollen in snow, firn and glacial ice are not fully understood and currently challenge the use of pollen in ice-core studies (Ewing et al., 2014). Just in few cases pollen has appeared as a potential dating material, when seasonal layers are preserved (Festi et al., 2017). Yet, pollen concentrates have been used in other type of archives with high success (Fletcher et al., 2017), opening the door to apply the same methodology here.

(iii) Two ice samples (MP-67 and MP-81), which appeared darker than others once melted, were filtered throughout a filtration line connected to a vacuum pump using quartz fiber filters (PALL tissuquarzt 2500QAT-UP), parameterized at controlled conditions (temperature: 22 – 24 °C; relative humidity 25 – 35 %) and weighted twice in different days. Abundant material was obtained, but no control was made on the composition and amount of organic material versus other types of input. The three concentrated pollen samples and the two filters were dated at the same $^{14}$C dating laboratory (Direct AMS, Seattle, USA) (Table 3).

(iv) Finally, two more samples were dated at the Laboratory of Environmental Chemistry, Paul Scherrer Institute (Switzerland) removing the outer part of the ice core segment for decontamination purposes (Jenk et al., 2009). Since organic fragments (plants, wood, insects) are rarely found in mountain glaciers, a complementary dating tool was recently developed based on extracting the microgram-amounts of the water-insoluble organic carbon (WIOC) fraction of carbonaceous aerosols embedded in the ice matrix for subsequent $^{14}$C dating (Uglietti et al., 2016). These two samples, labelled as MP10m and MP59m at the WIOC facility (Table 3), were selected as they were the only ones with sufficient ice volume available.

Once the 16 radiocarbon ages were obtained, we converted them into calendar ages by using the CALIB 5.0.2 software, which uses the most updated dataset, INTCAL13 (Reimer et al., 2013) (Table 3). The median of the one-$\sigma$ probability interval was selected for these dates, resulting in highly variable errors in the calendar ages obtained (from 30 years on the bulk organic samples to more than 200 years on pollen and WIOC samples). While the first method to select organic remains at the microscope resulted the best option, the pollen concentration and filtering methods

used to isolate organic matter to be dated by $^{14}$C were, unfortunately, not successful.
Finally, from the initial 16 dates, we had to discard seven according to the following
criteria (see the "comments" column in Table 3):
- Sample MP-46 (D-AMS 025295) was the only one discarded from the nine initial
bulk organic matter samples. We suspect that the very recent age obtained
(1897 ± 20 CE, Table 3) is due to the sample contamination, since small plastic
debris coming off from the painting used in the coring device were identified
under the microscope.
- From the two WIOC-dated samples, one was discarded (MP10m) due to the
low carbon content (5.3 μg), thus providing too inaccurate results (854 ± 721
CE, with an unacceptable large uncertainty). The other sample (MP59m), with
higher organic carbon content (28.7 μg), was incorporated into the age model
in spite of its error above 200 yr (1046 ± 242 CE).
- The three pollen concentrates provided unreliably old dates with very high
errors, likely due to the small amount of pollen that we were able to
concentrate (errors above 200 yr, Table 3). Obtaining old dates from pollen is a
quite common problem not yet solved in the literature (Kilian et al., 2002).
- Similarly, we discarded the two filter samples MP-67 and MP-81 (D-AMS
029894 and D-AMS 033972, respectively). The material accumulated in the
filters was a mixture of particles containing detrital carbonate eroded from
Eocene limestones or supplied by Saharan dust, which was not removed and
probably influenced the results incorporating allochthonous carbon to the
samples.
Finally, nine dates were employed to infer the chronology of the MPG sequence. The
depth–age model was created using a linear regression in the R package CLAM 2.2
(Blaauw, 2010; Blaauw et al., 2019).
*3.4. Trace elements in soluble and insoluble material.*
Thirty-five selected ice samples from the altitudinal transect were melted and filtered
through a filtration ramp connected to a vacuum pump using quartz fibre filters (PALL
tissuquarzt 2500QAT-UP). Filters were pre-heated at 250 °C and thereafter prepared in
controlled conditions (temperature: 22 – 24 °C; relative humidity: 25 – 35 %) before
and after filtration. Subsequently, they were weighted in two different days. Mass
difference between blank and sampled filters was used to calculate the amount of
insoluble material entrapped in ice samples. For every sample, an aliquot and a filter
were obtained. From aliquots, anions and cations, as well as major and trace elements
were determined. From filters, we determined major and trace elements, as well as
organic and elemental carbon, following the method devised by Pey et al. (2013)
(Table 4). Basically, an acidic digestion ($HNO_3$:$HF$:$HClO_4$) of half of each filter was
conducted, driven to complete dryness, being the remaining material re-dissolved in
$HNO_3$. Inductively coupled plasma mass spectrometry (ICP-MS) and inductively
coupled plasma atomic emission spectroscopy (ICP-AES) were used to determine major
and trace elements, respectively. From the other half of each filter, a 1.5 $cm^2$ section
was used to determine Organic Carbon (OC) and Elemental Carbon (EC) concentrations
by using a SUNSET thermo-optical analyzer, following the EUSAAR_2 temperature
protocol. Table 1 also contains the Enrichment Factors (EFs), calculated as follows:

$$EF_{iCODD} = \frac{X_{iCODD}/Al_{CODD}}{X_{iUC}/Al_{UC}} \quad ; \quad EF_{iMPGID} = \frac{X_{iMPGID}/Al_{MPGID}}{X_{iUC}/Al_{UC}} \quad ; \quad EF_i = \frac{X_{iCODD}/Al_{CODD}}{X_{iMPGID}/Al_{MPGID}}$$


where $EF_{iCODD}$ is the Al-normalised Enrichment Factor with respect to the Upper Crust
(*UC*, Taylor and McLennan, 1995)) of an *'i'* element in the current Ordesa's deposited
dust (*CODD*); $EF_{iMPGID}$ is the Al-normalised Enrichment Factor with respect to the *UC* of
an *'i'* element in the current MPG ice dust (*MPGID*); and $EF_i$ is the Al-normalised
Enrichment Factor with respect to *CODD* of an *'i'* element in the *MPGID*.
Regarding the Pb/Al ratio, we carried out a normalization with Al in both, ice and lake
records, to disentangle the anthropogenic lead variability from possible detrital inputs.
Aluminium has been selected for normalization since this lithogenic element is
immobile and abundant in carbonated watersheds (Corella et al., 2018).
*3.5. Hg determination*.
Total Hg concentration measurements were carried out on 21 selected samples by
Atomic Absorption Spectrophotometry using an Advance Mercury Analyzer (AMA 254,

LECO Company). This equipment is specifically designed for direct mercury determination in solid and liquid samples without sample chemical pre-treatment. Certified reference materials were used to determine the accuracy and precision of the Hg measurements. These reference materials were ZC73027 (rice, 4.8 ± 0.8 μg kg$^{-1}$) and CRM051–050 (clay soil, 4.08 ± 0.09 mg kg$^{-1}$). The standard deviation (repeatability) was ≤ 15 % and the relative uncertainty associated with the method (with a confidence level of about 95 %) was ± 20 %. All analyses were run at least three times. Total metal concentrations were expressed in μg g$^{-1}$ of dry weight sediment due to the low amount detected.

**4. Results**

*4.1. Chronological model*

To date the ice sequence from MPG we compiled the results from $^{137}$Cs, $^{210}$Pb and $^{14}$C methods. First, we note the that the all samples analyzed for $^{137}$Cs presented activities below the MDA values (Minimum Detection Activity) (Table 1). These values, compared to other $^{137}$Cs values in glacier records (e.g. Di Stefano et al., 2019), indicate that all samples are older than 60–65 years and therefore they were not exposed to the atmosphere after 1950 CE. Similarly, $^{210}$Pb activity was also undetectable in most cases, except in three samples (MP-100, MP-73 and MP-76) with concentrations above MDA (Table 2), but well below the usual $^{210}$Pb activity concentrations in glacier surface samples from European Alps, which are on average 86 ± 16 mBq kg$^{-1}$ (Gäggeler et al., 2020). These three samples contained a large amount of lithogenic particulate material from atmospheric dust or ash deposits, likely causing the observed values. Thus, the absence of $^{210}$Pb activity in the analysed samples suggests that MPG ice samples were very likely older than 100 years and the $^{210}$Pb had completely decayed. We then built up the proposed MPG chronology using only AMS $^{14}$C dating.

Regarding $^{14}$C dating, we took most of the ice samples for dating in sections where dark debris layers alternated every ca. 5 m with cleaner and clearer ice (Fig. 2). The debris-rich layers were composed of detrital, silty-sandy size deposits, likely coming from wind-blown particles (e.g. black carbon-rich particles, dust) and from erosive processes of the limestone catchment, including frost weathering and the fall of

gravel-sized particles from the surrounding cliffs. These debris-rich layers do not have a subglacial origin since they are observed all along the sample profile and large accumulation of debris, characteristic of subglacial glacier till, were not present at MPG. These debris layers contain more organic remains than those formed by clear ice, making them ideal spots to find datable remains.

Interestingly, the frequency of debris layers increases towards the top of the glacier sequence. We consider the accumulation of debris layers to be indicative of reduced ice accumulation and dominance of ablation periods. In such situations, the detrital and organic material concentrate as the ice melts, giving its characteristic dark colour to the ice layers. The major concentration of such layers occurred among samples MP-67 and MP-73 (Table 3), thus suggesting the dominance of ablation processes. Therefore, we run the depth–age model setting a hiatus at 73 m depth, where we infer an interruption in the ice accumulation was produced. Finally, as explained in Methods, the depth-age model was constructed with nine of the 16 initially dated samples (Table 3). Given the scattered depths at which dates concentrate, we chose to perform a non-smooth, linear regression for preventing any model over-fitting and a spurious depth-age relationship (Fig. 3). Details on how the model was performed and a reproducible workflow with the current chronological dataset are available at https://zenodo.org/record/3886911.

4.2. Trace elements

We have used the averaged concentration values of major and trace elements currently obtained at a monitoring station located at the Ordesa site (OMPNP; 8 km away from the MPG, at 1190 m a.s.l.), where deposited atmospheric particulate matter is sampled monthly (Table 4) (Pey et al., 2020). Interestingly, the elements that are abundant nowadays in the Ordesa station are not so frequent in the ice from MPG. Indicators such as organic carbon, Zn, Se and Cd concentrations, all of which are potential proxies of current anthropogenic emissions, are much higher in the samples from Ordesa, which are representative of today's atmosphere, than in the ice samples from the MPG. In fact, similar results appear when comparing with other glaciers in Europe where the EFs for some elements (eg. Zn, Ag, Bi, Sb and Cd) are well above the

crustal value (Gabrieli et al., 2011), demonstrating the predominance of non-crustal
deposits and suggesting an anthropogenic origin. The low concentration of those
elements in MPG samples could indicate their disappearance from the glacier surface
layers due to continuous melting. This supports our suggested depth-age model (Fig.
3), in which ages from the Industrial Period are not recorded. Conversely, the Al-
normalised enrichment factor (EF) of Ti, Mn, Cr, Co, Ni, Cu and Pb, elements linked to
the natural fraction (dust deposition, lithogenic elements) and mining activities
(Corella et al., 2018), are more abundant in the MPG ice samples than in the present-
day Ordesa aerosols (Table 4). From them, Cu and Pb were markedly enriched (by a
factor > 6) in the MPG ice samples compared with the current deposited aerosols in
Ordesa station.
**5. Discussion**
*5.1. Dating Monte Perdido Glacier ice sequence*
Dating ice from non-polar glaciers is challenging and often problematic as annual layer
counting is precluded due to periods without net accumulation, and to common ice
deformation caused by glacier flow (Bohleber, 2019; Festi et al., 2017). The low values
in $^{137}$C and $^{210}$Pb activities in MPG samples compared to other European glaciers (Di
Stefano et al., 2019; Festi et al., 2020; Gäggeler et al., 2020) do not allow building any
chronology for the last 150 years (Tables 1 and 2) and, therefore, we have constrained
the depth-age model of MPG ice using nine $^{14}$C absolute dates from different materials
(Table 3). We have also integrated in the chronology the characteristics of the ice
stratigraphy, such as the presence of dark debris-rich layers.
Our MPG depth-age model suggests that the glacier is composed of ice up to ~2000
years old, and that the glacier's subsequent history has involved three main periods
(Fig. 3). Period I was an accumulation phase from 0 to 700 CE. Period II represents an
ablation-dominated phase from 700 to 1200 CE, which corresponds to the dark-rich
layer interval where more dates are concentrated. Period III corresponds to a new
accumulation phase from 1200 to 1400 CE. This last period agrees well with an
increase in heavy rainfall events during the cold season (Oct-May) in the Southern
Central Pyrenees between 1164 – 1414 CE (Corella et al., 2016), which most likely

resulted in higher snow accumulation at high elevation areas, leading to a net accumulation on the MPG. Finally, no ice formed during, at least, the last 600 years in the MPG. This indicates that the LIA ice has been melted away, pointing to a period of intense mass loss since 1850 CE. The MPG age model is supported by, first, a quantitative comparison with present-day atmospheric particulate matter (Table 4) and, second, by the comparison with the paleoenvironmental sequence of the Marboré Lake for the last 2000 years (Corella et al., 2021; Oliva-Urcia et al., 2018) (Fig. 4).

Present-day aerosols in the studied region are well-recorded at the nearby Ordesa site (Pey et al., 2020). Following previous studies on present-day atmospheric particulate matter composition from natural, urban or industrial areas (Querol et al., 2007), the values of some elemental ratios (e.g., Cu/Mn, As/Se, Pb/Zn) help to determine the origin of the particulate matter accumulated today. The Ordesa site can accordingly be mostly defined as remote in terms of atmospheric deposition ("rural background") while the average composition of MPG ice samples could be defined as a site under the influence of Cu mining and smelting activities, due to the high values of the Cu/Mn, As/Se and Pb/Zn ratios. It is noteworthy that Cu, Ag, and Pb mining and smelting have been historically documented in Bielsa valley during pre-industrial times (Callén, 1996). Indeed, MPG is only 7 km east from some of the largest lead and silver ore deposits in the Central Pyrenees (historical mines of Parzán). The impact of ancient environmental pollution in high alpine environments is archived in the lacustrine sequence of the neighbouring Marboré Lake, providing first evidences of long-range transport of trace metals from historical metal mining and smelting activities during the Roman Period (RP) (Corella et al., 2018, 2021). Similar ice core records from the Alps have also demonstrated the suitability of glacier ice to record local and regional mining and smelting activities during RP and pre-Roman times (More et al., 2017; Preunkert et al., 2019). Even if the enrichment of trace elements in the MPG ice record may correspond to mining activities during ancient times, the distinct elevation of MPG with respect to Alpine glaciers where such activities were recorded (> 4000m a.s.l.), together with the likely processes of redistribution of chemical impurities due to percolation (Pohjola et al., 2002), prevents a firm interpretation of the origin of these elements.

On the other hand, the comparison of Pb/Al ratios from the independently dated records of Marboré Lake and MPG provides further support for our MPG chronology (Fig. 4). In particular, the lack of a Pb/Al peak characterizing the Industrial Period in the upper sequence of the MPG record, where several samples were analysed (see ID in Table 5) supports the absence of records from the last two centuries in the MPG, in agreement with the results of $^{210}$Pb and $^{137}$Cs analyses. Similarly, the Hg concentration in the glacier is uniform throughout the ice sequence (Fig.4). Concentrations of Hg in other ice core records show an increase during the onset of Industrialization at 1800 CE with maximum values typically 3–10 times higher than preindustrial values (Cooke et al., 2020). In the Marboré Lake, the Hg increase occurred over the last 500 years associated to the maximum activity in the Spanish Almadén mines during the Colonial Period (Corella et al., 2021). Again, these results, lacking an expected increase in Hg levels, support the depth-age model for the MPG record where the last six centuries of ice deposition are missing.

*5.2. Evolution of the Monte Perdido glacier over the last 2000 years*

The analysed ice from MPG provides valuable information about the evolution of the glacier over the last two millennia, which deserves consideration in the regional context. Based on published results, the oldest paleoclimatic information in the Marboré Cirque comes from the Marboré Lake, since no glacier deposits corresponding to the Late Pleistocene have been found in the cirque (García-Ruiz et al., 2014). There is sedimentological evidence that the Marboré Lake was already ice-free at least since the onset of the Bølling period (Greenland Interstadial-1; 14,600–12,900 yr BP), when clastic sediments were deposited in the lake basin (Leunda et al., 2017; Oliva-Urcia et al., 2018). This is coherent with the nearby La Larri glaciolacustrine sequence which showed that the main Pineta Glacier had already retreated further up in the headwater by 11 kyr BP (Salazar et al., 2013). In fact, glaciological studies performed in the Central Pyrenees confirm the sudden retreat of glaciers during the Bølling period, when they were reduced to small ice tongues or cirque glaciers (Palacios et al., 2017). The next piece of information comes from the outermost moraine that was dated at 6900 ± 800 $^{36}$Cl yr BP (García-Ruiz et al., 2020), corresponding to the Neoglacial advance, a cold period identified in the sediments of

Marboré Lake (Leunda et al., 2017). Other minor advances would have occurred in the
MPG prior to the LIA, as inferred from three polished surfaces dated at 3500 ± 400,
2500 ± 300 and 1100 ± 100 [36]Cl yr BP (García-Ruiz et al., 2020).
With the new chronology of the MPG record, we can ascertain that the MPG has
persisted at least since the RP (ca. 2000 years ago). At that time, which is a well-known
warm period in the Iberian Peninsula as recorded in both continental (Martín-Puertas
et al., 2010; Morellón et al., 2009) and marine sequences (Cisneros et al., 2016;
Margaritelli et al., 2020), the glacier was still present, but probably smaller than during
previous Neoglacial times (Figs. 5A and 5B). This situation probably continued during
the following cold period, the Dark Ages (DA, Fig 5C) when the glacier advanced as
indicated by the polished surface dated at 1100 ± 100 [36]Cl yr BP (García-Ruiz et al.,
2020). In the Alps, reconstructions based on dating trees found within and at the edge
of glacier forefields have revealed a minimum glacier extent during the Iron Age and
the RP (Holzhauser et al., 2005), when glaciers were estimated to be smaller than
during the 1920s (Ivy-Ochs et al., 2009). Afterwards, in the late RP and the early Middle
Ages numerous glaciers in the Alps advanced during the DA, also known as the
Göschener II oscillation (Holzhauser et al., 2005).
The Medieval Climate Anomaly (MCA, 900–1300 CE) is the most recent preindustrial
warm era in Europe (Mann et al., 2009). For instance, in the Alps, a general glacier
retreat has been observed during this period, mainly associated with a decline in
precipitation (Holzhauser et al., 2005). According to the depth-age model, the MPG
experienced a dramatic retreat during that period (Fig. 5D), including the complete
melting of some minor glaciers in the Marboré Cirque (García-Ruiz et al., 2020).
Nevertheless, during the MCA part of MPG was preserved, as we find ice from 0 to 700
CE. No doubt the ice loss was significant, as evidenced by the accumulation of dark
strata over a long time interval (600 – 1200 CE) and the just six meters of ice remaining
from that period (blue horizontal line, Fig.3). On this basis, we propose that the MPG
was dominated by ablation processes during the MCA. It is evident that at the end of
the MCA the MPG still preserved ice from the RP and the first half of the DA (Fig. 5D). It
is difficult to confirm if Neoglacial basal ice is still present in MPG since no ice sample
was dated with Neoglacial age or even older. Still, Neoglacial ice could have remained

in the glacier base without being exposed by the slope where sampling procedures were carried out.

Over such a diminished MCA glacier, ice started to accumulate again at a rapid rate during the LIA (1300 – 1850 CE). In most cases, the LIA was the period when mountain glaciers recorded their maximum Holocene extent (Solomina et al., 2016), with remarkable advances in Alpine glaciers (Ivy-Ochs et al., 2009). From a large variety of proxies, several warm and cold periods have been identified in the Iberian Peninsula during the LIA (Oliva et al., 2018). In the Marboré Cirque two generations of LIA moraines have been mapped (García-Ruiz et al., 2014), whose emplacement coincided with the coldest LIA phases, i.e. 1620 - 1715 CE, when the Pyrenean glaciers recorded their maximum extent of the last two millennia, and at some time between 1820 - 1840 CE, when a rapid advance of the ice mass moved over the large moraine leaving parallel ridges and furrows, so-called flutes, as signs of erosion (García-Ruiz et al., 2020; Serrano and Martín-Moreno, 2018). These two cold phases are very well identified in the Marboré Cirque and were confirmed by the study of the altitudinal fluctuations of the timberline in the neighboring Escuaín Valley (Camarero et al., 2015). In fact, according to the map of Schrader from 1874 CE and other historical sources, the MPG made direct contact with the large moraine in the second half of the 19[th] century (García-Ruiz et al., 2014). Despite the fact that the MPG would have covered an area of 5.56 km$^2$ at the end of the LIA (Fig. 5E) (González Trueba et al., 2008), there is no record today of ice accumulated during the LIA, except for a few meters at the top of the sequence corresponding to about 1400 CE. This means that more than 600 years of ice accumulation have been lost associated with warming after ca. 1850 CE. This situation is not so common in the Alps, where ice from the LIA, and even from the last two centuries, is still commonly preserved in many studied cold glaciers (Eichler et al., 2000; Gabrielli et al., 2016; Gäggeler et al., 1983; Preunkert et al., 2019).

Today the MPG is divided in two small ice bodies that together cover just 0.38 km$^2$ (López-Moreno et al., 2016; Fig. 5F). Comparing the MPG extent at the end of the LIA (ca. 1850 CE), as given by the moraine location, and today's extent, more than 5 km$^2$ of MPG has disappeared indicating that the last 150 years have likely been the period with the largest glacier melting over the last 2000 years.

**5. Conclusions**

This study presents for the first time a continuous chronological model of a remaining small glacier in the Pyrenees, reconstructed from a set of $^{14}$C dates on different organic remains, and supported by measurements of current atmospheric deposition and comparison with a nearby lake sequence (Marboré Lake). The ice sequence from the Monte Perdido Glacier (MPG) covers the last 2000 years, allowing the identification of cold time periods of glacier growth and warm time periods of ice loss. We demonstrate that the glacier was active during the Roman Period (RP), a well-known warm period in the Iberian Peninsula. During the Medieval Climate Anomaly (MCA), the MPG experienced a dramatic retreat marked by the presence of dark debris layers interpreted in terms of successive years when ablation processes predominated. The Little Ice Age (LIA) was a period of glacier growth, but it is not recorded today in the ice from MPG, since more than 600 years of ice accumulation have been lost associated to the warming after the end of the LIA, at ca. 1850 CE. This evidence from the depth-age model is supported by the lack of anthropogenic indicators usually associated with the Industrial Era, which are abundant today in the current atmospheric deposition in a nearby site. Additionally, both the Hg concentration and the Pb/Al ratio appear much higher in the Marboré Lake sediments, whereas the MPG record does not reflect their anthropogenic increase.

Comparing the present-day glacier situation with that of previous warm intervals, such as the RP or the MCA, we conclude that the MPG is nowadays greatly reduced in area and volume. Additionally, the recent rate of ice-mass loss is definitely more rapid than that of the four centuries spanned by the MCA, thus suggesting that present day warming in the Pyrenees is faster and more intense than in any previous warm phase of the last 2000 years. Under the current climatic conditions, it is reasonable to expect the disappearance of this glacier, as well as other glaciers in the Pyrenees and in Southern Europe, over the next few decades.

**6. Data availability**

The input data file for CLAM as well as the output results are stored in the open repository Zenodo (https://zenodo.org/record/3886911). The rest of data are given in the paper Tables.

**7. Author contributions**

The paper was conceived by A.M., M.B., C.S. and J.I.L-M. together with F.N., J.O-G., J.L., P.G-S., C.C., J.L-M., B.O-U, S.H.F and J.G-R. who contributed to design and develop this research project (PaleoICE). F.N., C.P., M.L., E.A. participated during field work to recover the samples; A.M., M.B. and M.L. prepared the samples for $^{14}$C dating; J.G.O. carried out the $^{210}$Pb and $^{137}$Cs analyses; J.P., X.Q. and A.A. provided the geochemical data from Ordesa site and MPG; J.P.C., M.J.S. and R.M. provided the Hg data from Marboré Lake and MPG; and G.G.-R. built the age depth-model. All authors contributed to discuss and interpret the data and to the writing of the original and revised version of this paper.

**8. Competing interest**

The authors declare that they have no conflict of interest.

**9. Acknowledgements**

The Spanish Agencia Estatal de Investigación (AEI – Spain) and the European Funds for Regional Development (FEDER – European Union) are gratefully acknowledged for financial support through PaleoICE EXPLORA project (CGL2015-72167-EXP), CGL2015-68993-R, CGL2015-69160-R and CTM2017-84441-R projects (AEI/FEDER, UE) and through the iMechPro RETOS project (RTI2018-100696-B-I00). S.H.F. and J.G.-O. acknowledge support by the Spanish Government through María de Maeztu excellence accreditation 2018-2022 ref MDM-2017-0714 and ref CEX-2019-000940-M, respectively. M.B. is supported by postdoctoral fellowship Juan de la Cierva-Formación program provided by the Spanish Ministry of Science, Innovation and Universities (ref.: FJCI-2017-34235063753). The authors are grateful to Eduardo Bartolomé and José Estebán Lozano for their help manufacturing parts of the coring devices and to the support provided by the Dirección General de Conservación del Medio Natural (Government of Aragón) and by the staff of the Ordesa and Monte Perdido National

Park during our field campaigns. This study contributes to the work carried out by the
GA research group Procesos Geoambientales y Cambio Global (ref E02-20R) and MERS
research group 2017 SGR 1588.

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

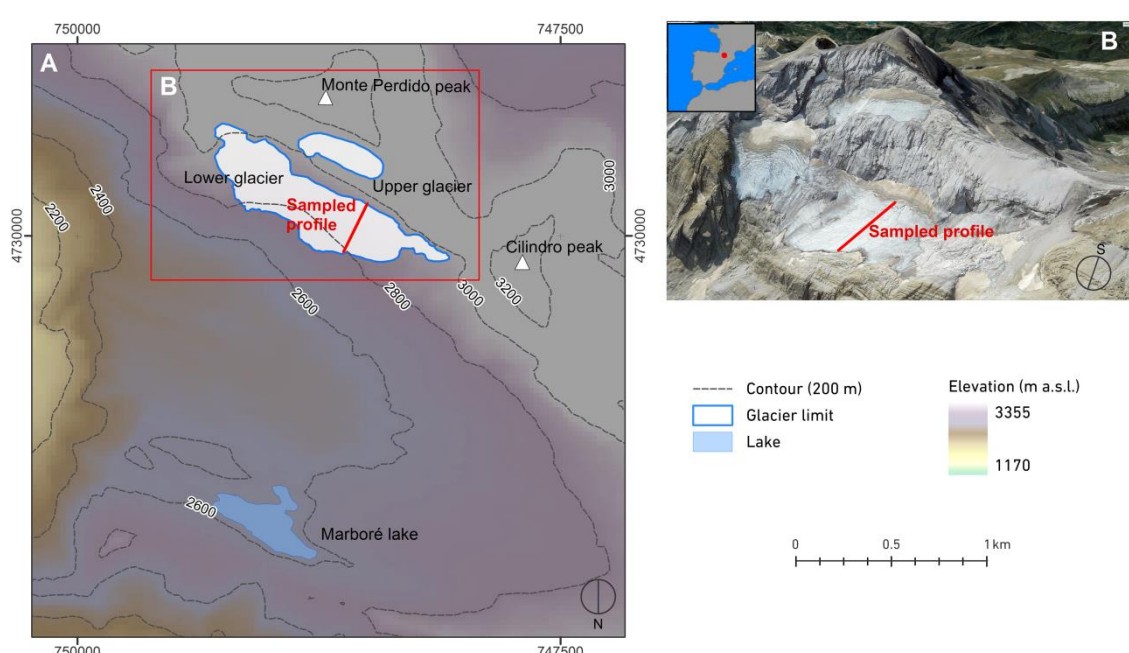


**Figure 1**. (A) Location of Monte Perdido Glacier (MPG) within a digital elevation map of
Marboré Cirque; (B) Picture (©Google Earth) of MPG where the location of the
sampled profile is indicated (see Fig. 2).


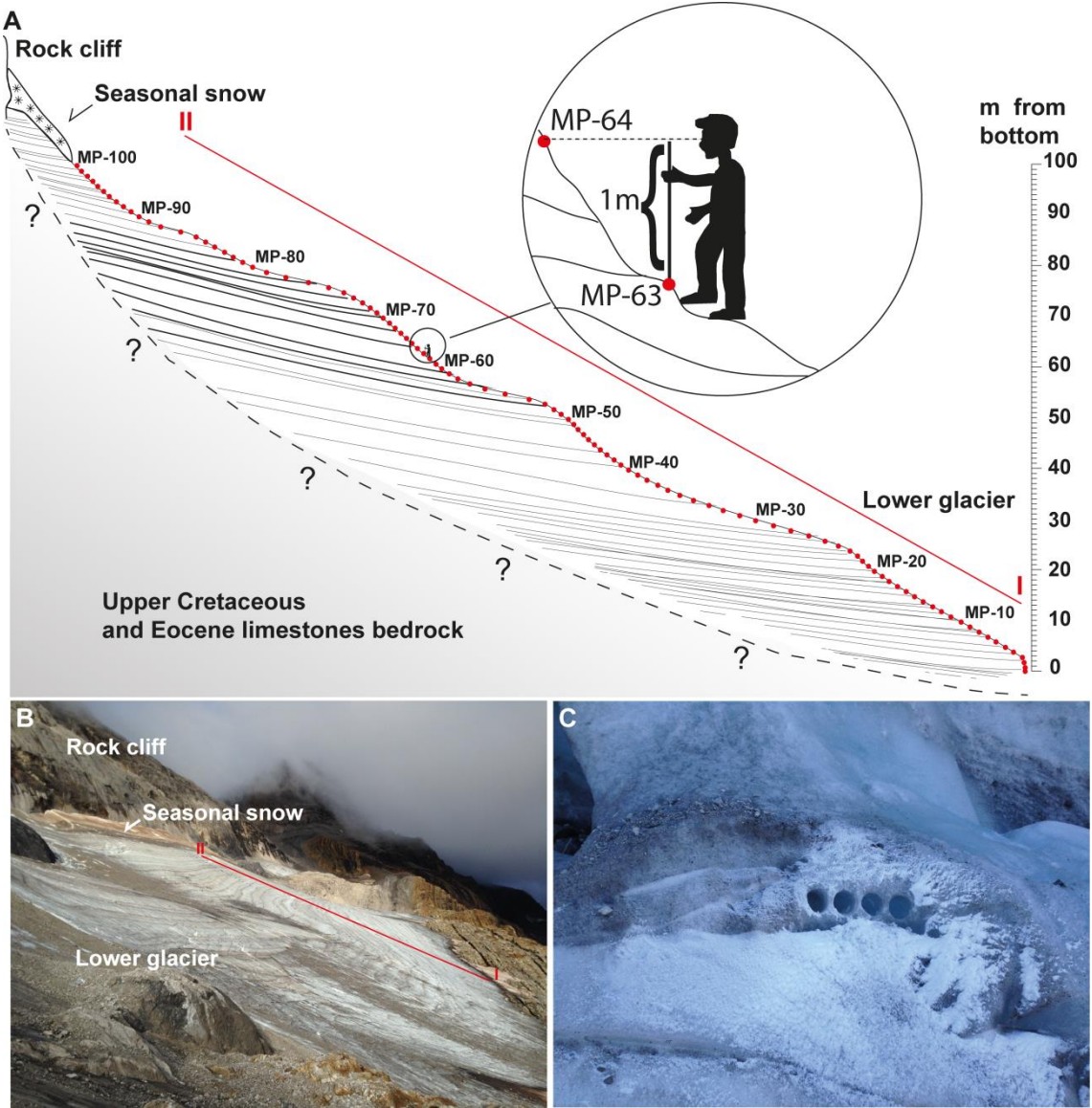


**Figure 2**. (A) Simplified scheme with the position of the 100 samples collected along
the slope (red line I-II marks the profile indicated in B; identification of the samples is
MP-0 to MP-100). According to the ice bedding (tilt is approximate) the oldest material
should be found at the bottom of the lower glacier. The number of glacier layers is
drawn according to the layers observed in the image depicted in (B). Note the inset
with a detailed view of the sampling procedure measuring a height difference of 1 m to
obtain every sample. (B) Image of the Monte Perdido glacier surface where the
sampling was carried out (red line I- II represents the sampled profile shown in Figure
1). Note the presence of dark debris-rich layers alternating with cleaner ice. (C).
Detailed view indicating that every sample consisted in 3-4 small horizontally-drilled
cylinders (see text for more details).

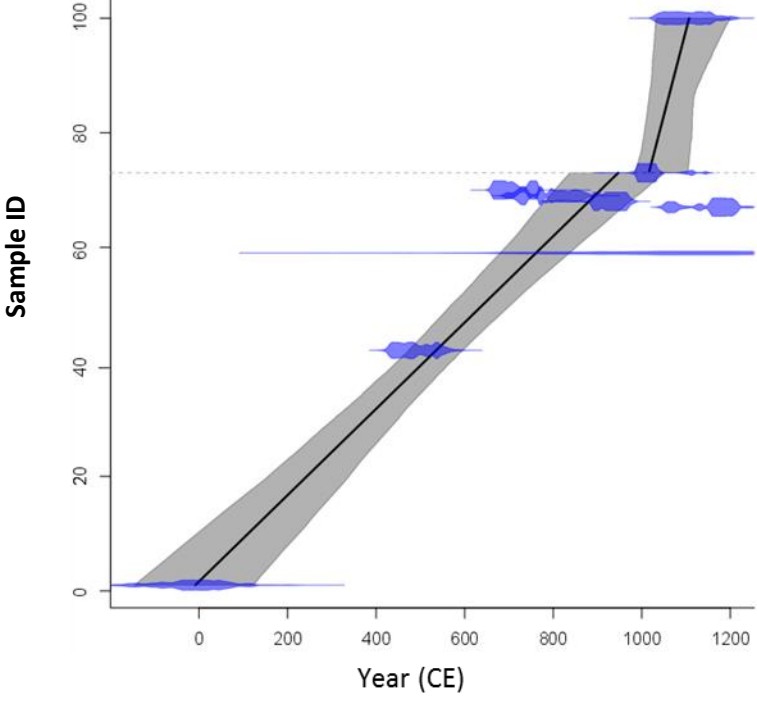


**Figure 3**. Age model for the Monte Perdido ice sequence based on linear interpolation
of [14]C data (Table 3), obtained using the Clam software (Blaauw, 2010; Blaauw et al.,
2019). Y axis indicates the number of samples from MP-0 to MP-100 (see Fig. 2). The
dates appear as the calendar-age probability distributions in blue, while the black line
is the resulting depth-age model and the gray envelope shows the 95 % confidence
interval. Note the hiatus located at 73 m indicated by a dashed line. The error of
sample MP59m is so high that appears as a horizontal line.

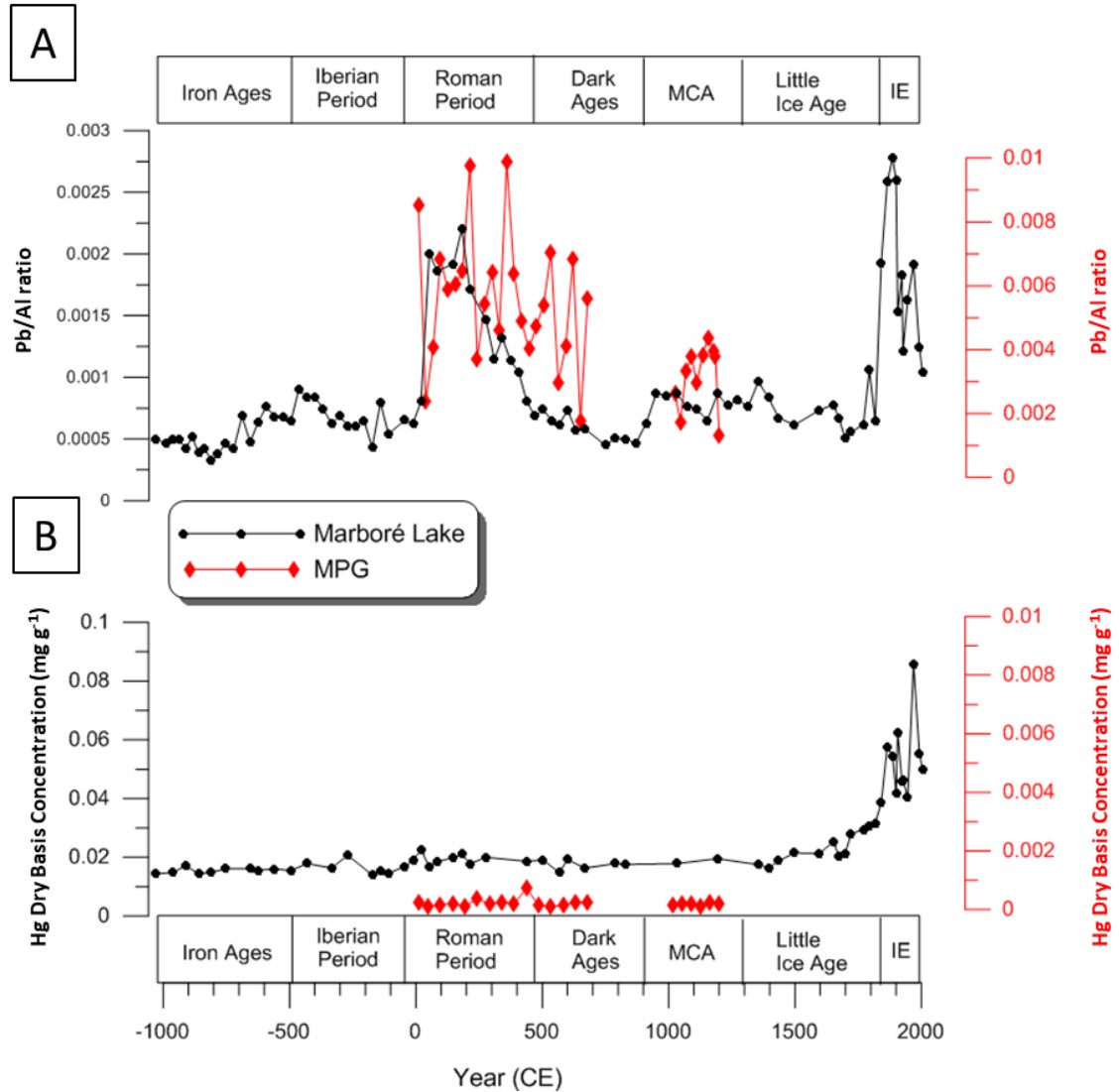

**Figure 4.** Comparison of Pb/Al ratio and Hg concentration of dry weight sediment in
MPG samples with data obtained from Marboré Lake sediments (Corella et al., 2021).
Note the differences in the vertical axis. Sample IDs from MPG are indicated in Table 5.

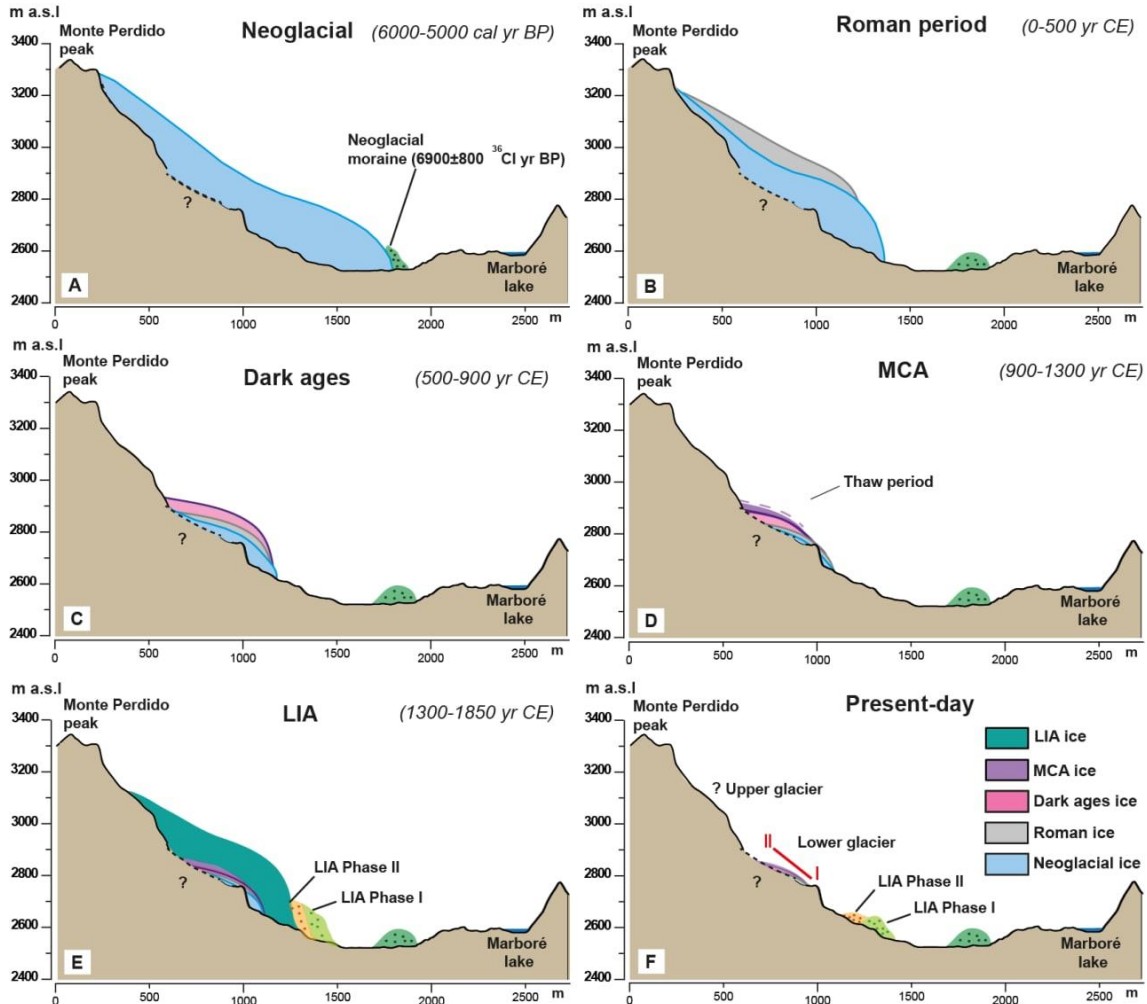


**Figure 5.** Schematic geomorphic transects (south to north) taken from the Marboré Cirque, showing the tentative reconstruction of MPG during the six main stages discussed in the text. A) Neoglacial Period (ca. 5000 – 6000 cal yr BP) where the Neoglacial moraine is indicated (García-Ruiz et al., 2014). This figure represents the state of maximum glacier advance during the Neoglacial period. (B) Roman Period (0-500 CE) when the glacier is shown considerably retreated. (C) Dark Ages (500-900 CE). (D) Medieval Climate Anomaly (900-1300 CE), a period when the glacier retreated and ablation caused a concentration of debris and organic remains form dark layers in the glacier ice (discontinuous line aims to highlight the importance of melting processes). (E) Little Ice Age (1300-1850 CE), with the MPG reaching the LIA moraines position, thus represented at its maximum advance during that period. (F) Present-day situation characterized by the MPG divided into two ice bodies, no ice remaining from the LIA, and very steep slopes (sampling transect indicated by a red line).

**Table 1.** Concentrations of $^{137}$Cs in the soluble water fraction of ice from Monte
Perdido samples. MDA: Minimum Detection Activity

| Sample | Mass of ice analyzed (g) | $^{137}$Cs activity (Bq·L$^{-1}$) | MDA (Bq·L$^{-1}$) |
|---|---|---|---|
| MP-61 | 240 | < MDA | 0.15 |
| MP-82 | 178 | < MDA | 0.16 |
| MP-97 | 232 | < MDA | 0.14 |
| MP-98 | 376 | < MDA | 0.09 |
| MP-100 | 238 | < MDA | 0.17 |



**Table 2.** Determination of $^{210}$Pb activity in the soluble water fraction of 100 g of ice
from Monte Perdido samples. MDA: Minimum Detection Activity.

| Sample | $^{210}$Pb activity (mBq·L$^{-1}$) | MDA (mBq·L$^{-1}$) |
|---|---|---|
| MP-73 | 17.4 ± 2.6 | 1.14 |
| MP-76 | 6.2 ± 1.3 | 0.70 |
| MP-82 | <MDA | 0.61 |
| MP-85 | <MDA | 0.84 |
| MP-88 | <MDA | 1.23 |
| MP-91 | <MDA | 1.05 |
| MP-94 | <MDA | 0.71 |
| MP-97 | <MDA | 0.77 |
| MP-98 | <MDA | 0.58 |
| MP-100 | 8.5 ± 1.5 | 0.71 |



**Table 3.** Radiocarbon dating of MPG samples indicating their origin, the radiocarbon
age ($^{14}$C age BP) and the calibrated date using INTCAL13 curve and presented in
calendar years Common Era (CE). Samples in red *italics* were not included in the depth-
age model (see column "comments" and text for explanation).

| Sample origin | Sample ID | Laboratory ID | $^{14}$C age BP | Cal age (CE) | Comments |
|---|---|---|---|---|---|
| Bulk organic matter | MP-1 | D-AMS 025291 | 2000±64 | 8±66 | Used in the age model |
| | MP-42 | D-AMS 025294 | 1554±27 | 462±32 | Used in the age model |
| | MP-48 | D-AMS 025295 | 73±33 | *1897±20* | *Discarded due to plastic contamination* |
| | MP-67 | D-AMS 025296 | 876±29 | 1185±31 | Used in the age model |
| | MP-68 | D-AMS 026592 | 1128±22 | 942±24 | Used in the age model |
| | MP-69 | D-AMS 026593 | 1230±23 | 730±14 | Used in the age model |
| | MP-70 | D-AMS 025297 | 1308±28 | 680±16 | Used in the age model |
| | MP-73 | D-AMS 025298 | 1011±25 | 1012±16 | Used in the age model |
| | MP-100 | D-AMS 025299 | 923±39 | 1074±31 | Used in the age model |
| Bulk material (filter) | MP-67filter | D-AMS 029894 | 485±40 | *1429±15* | *Discarded due to mixing with detrital fraction* |
| | MP-81filter | D-AMS 033972 | 1758±25 | *287±68* | |
| WIOC | MP10m | MP10m | 812±755 | *854±721* | *Discarded due to too high error* |
| | MP59m | MP59m | 926±268 | 1046±242 | Used in the age model |
| Pollen concentration | MP-30pollen | D-AMS 031464 | 3906±42 | *-2384±1332* | *Discarded due to technical issues and too high errors* |
| | MP-70pollen | D-AMS 031465 | 1787±37 | *237±255* | |
| | MP-100pollen | D-AMS 031466 | 1854±30 | *158±807* | |


**Table 4.** Elemental concentration (ppm) of major and trace metals in both Ordesa's current deposited dust and MPG ice deposits (averaged values for the 35 analyzed samples), as well as Upper Crust (UC) elemental contents for comparison (Taylor and McLennan, 1995). On the right side, Al-normalised Enrichment Factors (EF) for dust components and elements for: $EF_i$, the MPG ice dust versus the current Ordesa's deposited dust (CODD); $EF_{iCODD}$, the CODD versus the UC; and $EF_{iMPGID}$, the MPG ice dust versus the UC. Numbers in bold italics in the EF represent anomalous values (elements enriched in Ordesa samples or in MPG ones).

| | Ordesa 2016-2017 (2-year atmospheric deposition) | | | Monte Perdido (ice dust: 35 filter samples) | | | Upper Crust | Al-Normalised Enrichment Factors | | |
|---|---|---|---|---|---|---|---|---|---|---|
| | Max | Min (ppm) | Average | Max | Min (ppm) | Average | (ppm) | $EF_i$ | $EF_{iCODD}$ | $EF_{iMPGID}$ |
| OC | 443270 | 49659 | 206814 | 436343 | 14793 | 126381 | | 0.4 | | |
| EC | 114519 | 12506 | 39995 | 112769 | 14668 | 40605 | | 0.6 | | |
| Al | 122401 | 7883 | 60410 | 506467 | 19611 | 98808 | 80400 | 1.0 | 1.0 | 1.0 |
| Ca | 22578 | 3182 | 9663 | 119648 | 256.7 | 11984 | 30000 | 0.8 | 0.4 | 0.3 |
| Fe | 63218 | 2901 | 32665 | 183957 | 12504 | 59477 | 35000 | 1.1 | 1.2 | 1.4 |
| K | 27478 | 3907 | 14839 | 57038 | 4001 | 18505 | 28000 | 0.8 | 0.7 | 0.5 |
| Mg | 27286 | 2105 | 12265 | 72210 | 3513 | 16645 | 13300 | 0.8 | 1.2 | 1.0 |
| Na | 5380 | 1.2 | 1413 | 25750 | 593 | 5126 | 28900 | 2.2 | 0.1 | 0.1 |
| Ti | 5035 | 257 | 2334 | 52192 | 3243 | 13662 | 3000 | **3.6** | 1.0 | **3.7** |
| Mn | 1656 | 128 | 582 | 3835 | 174 | 979 | 600 | 1.0 | 1.3 | 1.3 |
| Sr | 170 | 19 | 78 | 200 | 20 | 80 | 350 | 0.6 | 0.3 | 0.2 |
| Be | 7 | 0 | 2.1 | 2.3 | 0 | 0.4 | 3 | 0.1 | 0.9 | 0.1 |
| V | 208 | 10 | 76 | 257 | 28 | 107 | 60 | 0.9 | 1.7 | 1.5 |
| Cr | 720 | 5 | 118 | 2915 | 12 | 441 | 35 | **2.3** | **4.5** | **10.3** |
| Co | 32 | 0 | 7.6 | 49 | 5.4 | 20 | 10 | 1.6 | 1.0 | 1.6 |
| Ni | 414 | 7 | 55 | 1046 | 4.3 | 228 | 20 | **2.5** | **3.6** | **9.3** |
| Cu | 683 | 33 | 127 | 26451 | 92 | 3786 | 25 | **18.3** | **6.7** | **123.2** |
| Zn | 9391 | 164 | 1316 | 3826 | 171 | 988 | 71 | 0.5 | **24.7** | **11.3** |
| As | 26 | 2 | 10 | 51 | 5.3 | 18 | 1.5 | 1.0 | **9.1** | **9.6** |
| Se | 90 | 0 | 22 | 30 | 0 | 5.2 | 50 | 0.1 | 0.6 | 0.1 |
| Cd | 100 | 0 | 14 | 1.5 | 0 | 0.3 | 0.98 | 0.0 | **18.8** | 0.2 |
| Sb | 26 | 0 | 4.5 | 59 | 2 | 11 | 0.2 | 1.5 | **29.7** | **43.3** |
| Ba | 1010 | 15 | 287 | 870 | 67 | 317 | 550 | 0.7 | 0.7 | 0.5 |
| Tl | 1 | 0 | 0.1 | 1.1 | 0 | 0.2 | 0.75 | 1.7 | 0.1 | 0.2 |
| Pb | 175 | 8 | 53 | 2989 | 86 | 495 | 17 | **5.7** | **4.2** | **23.7** |
| Th | 37 | 1 | 12 | 26 | 1.6 | 9.7 | 10.7 | 0.5 | 1.5 | 0.7 |
| U | 8 | 0 | 2.5 | 15 | 0 | 3.7 | 2.8 | 0.9 | 1.2 | 1.1 |

890 **Table 5**. Values of Pb/Al ratio and Hg concentration from MPG samples (plotted in Fig.

891 4).

| Pb/Al ratio in MPG | | | Hg in MPG | | |
|---|---|---|---|---|---|
| Sample ID | Age (yr AD) | Pb/Al | Sample ID | Age (yr AD) | Hg (μg/g) |
| MP-1 | 9.7 | 0.0085 | MP-1 | 9.7 | 0.00023 |
| MP-4 | 38.9 | 0.0024 | MP-5 | 48.6 | 0.00010 |
| MP-7 | 68.0 | 0.0041 | MP-10 | 97.1 | 0.00017 |
| MP-10 | 97.1 | 0.0068 | MP-15 | 145.7 | 0.00021 |
| MP-13 | 126.3 | 0.0059 | MP-20 | 194.3 | 0.00012 |
| MP-16 | 155.4 | 0.0060 | MP-25 | 242.9 | 0.00037 |
| MP-19 | 184.6 | 0.0065 | MP-30 | 291.4 | 0.00018 |
| MP-22 | 213.7 | 0.0098 | MP-35 | 340.0 | 0.00026 |
| MP-25 | 242.9 | 0.0037 | MP-40 | 388.6 | 0.00019 |
| MP-28 | 272.0 | 0.0054 | MP-45 | 437.1 | 0.00073 |
| MP-31 | 301.1 | 0.0064 | MP-50 | 485.7 | 0.00014 |
| MP-34 | 330.3 | 0.0046 | MP-55 | 534.3 | 0.00009 |
| MP-37 | 359.4 | 0.0099 | MP-60 | 582.9 | 0.00015 |
| MP-40 | 388.6 | 0.0064 | MP-65 | 631.4 | 0.00024 |
| MP-43 | 417.7 | 0.0049 | MP-70 | 680.0 | 0.00024 |
| MP-46 | 446.9 | 0.0040 | MP-75 | 1017.3 | 0.00014 |
| MP-49 | 476.0 | 0.0047 | MP-80 | 1053.8 | 0.00022 |
| MP-52 | 505.1 | 0.0054 | MP-85 | 1090.4 | 0.00019 |
| MP-55 | 534.3 | 0.0071 | MP-90 | 1126.9 | 0.00013 |
| MP-58 | 563.4 | 0.0030 | MP-95 | 1163.5 | 0.00023 |
| MP-61 | 592.6 | 0.0041 | MP-100 | 1200.0 | 0.00021 |
| MP-64 | 621.7 | 0.0068 | | | |
| MP-67 | 650.9 | 0.0018 | | | |
| MP-70 | 680.0 | 0.0056 | | | |
| MP-76 | 1024.6 | 0.0026 | | | |
| MP-79 | 1046.5 | 0.0017 | | | |
| MP-82 | 1068.5 | 0.0033 | | | |
| MP-85 | 1090.4 | 0.0038 | | | |
| MP-88 | 1112.3 | 0.0030 | | | |
| MP-91 | 1134.2 | 0.0039 | | | |
| MP-94 | 1156.2 | 0.0044 | | | |
| MP-97 | 1178.1 | 0.0040 | | | |
| MP-98 | 1185.4 | 0.0038 | | | |
| MP-100 | 1200.0 | 0.0013 | | | |

892