# Peer review of "The case of a southern European glacier which survived"

_The Cryosphere, 2020_

## Referee Comment (RC1) · Anonymous Referee #1 · 30 Jun 2020

General comments The paper is interesting and reports worthwhile chronological and elemental data, interpreted in terms of recent intense ablation removing ∼600 years of ice from MPG (despite the glacier surviving warm times before then). Unfortunately, as presented, I am not convinced the data analysis supports the main conclusions and I cannot recommend the manuscript be published in its current form.

First and most importantly, I believe the chronology needs to be addressed with greater structure and formal rigour. For example:

• Since this is so central to the paper's message, I find the translation from lateral surface samples to depth too difficult to follow in detail.

[Figure]

• It appears seven of 17 age samples were dismissed from the analysis; these need detailed comment on each.

• The interpretation of debris-rich englacial layers as periods of ablation (concentrating the debris) needs a far more rigorous argument based on physical analysis and exclusion of alternative possibilities. At present, the reader does not know whether these are isochronous, or whether they deform passively or cut-across primary layering/stratification. Could they be basally-derived? How are supply-rate variations excluded?

• How were the three core samples (and two deep samples) combined with the ∼100 surface samples?

• Since the annual 0 deg. C contour is at ∼3,000 m why wasn't snow/firn/ice sampled from the upper glacier. This glacier seems all to lie above this elevation and may therefore be accumulating, providing an undisturbed record of accumulation change?

• For me, too much key material seems to be given in the supplementary information rather than forming a central part of the argument of the main paper.

• For comparison, what elemental values have been measured in recent ice (not snow/firn)?

Second, substantial relevant material relating to the chronology of MPG appears later in the manuscript when I believe it should be summarised in full in the Introduction, directing the specific aim and objectives of this study. In my view, it's fine to introduce related information in the Discussion (such as the chronology of lakes in the area or of broader Alpine glaciers), but material directly relating to MPG - and particularly the focus of this submission (i.e., its chronology) - should form relevant background material in the Introduction.

Third, I found several of the sections and their contents to be confusing. For example, what I consider to be results are presented in the Field Site and Methods section, and

I would separate Results from Discussion.

Specific comments

Note, the manuscript includes many slight grammatical and typographical errors, as well as a few stylistic imperfections, which I have mostly not corrected. For me, the meaning was almost always clear, if not grammatically perfect.

Line/Location Comment/Suggestion

1 Title is awkward because 'that' is unspecified.

47 I'd avoid 'proves' unless the sentence states 'if we assume the chronological model is correct. . .'.

54-55 This claim needs to be supported by a rigorous analysis.

94-105 I'd invert this: state the aim of the paper and then state how it was achieved.

141 Data need to be presented and analysed to substantiate the claim that '(GPR and modelling). . . suggested that the oldest ice could be located in these areas.' This claim is central to the chronology presented in the manuscript.

161 1 m of what? A Jacob's staff should be described, or just use 'staff'.

175 Which 'uppermost five samples'?

175-6 These are Results

176-8 This is Interpretation

186-96 These are a mixture of Results and Interpretation. What values would be expected or have been measured on accumulating glaciers in similar settings?

209 How and where did these samples come from? How were they treated?

231-2 Why a second-order polynomial/quadratic? If there is a theoretical basis for such a relationship, then that should be presented. If there is not, then what justification is

there for this form?

234-8 This argument relating to debris concentration by ablation/low accumulation needs substantial background and argument, including arguments for this interpretation and against alternative interpretations of englacial debris bands. For example, how are supply-rate variations or a source at the glacier bed excluded?

288-91 These arguments need formalising, expanding and presenting as a logical progression of argument; currently, it is difficult to evaluate the accuracy of the chronology because this logical progression in argument (supported by illustrated data) is missing or split up through the paper and supplementary information.

281- I find this section difficult to follow since in some places Results have already been presented and they are only referred to here (e.g., 306-8) or Results are mixed with Interpretation, and in some case Methods are included here (318-27). In relating to the last point, I'd discuss sample removal from analysis in Methods and not Results.

328-30 These criteria should be discussed in full and presented in Methods rather than Results.

333 I'd bullet or number these three main periods and interpret consistent/repeated data across all three.

361- I'd move much of this into a dedicated Discussion section

412-26, 434-7 & 467-70 I'd move this published material relating specifically to the chronology of MPG to the Introduction. That way it would contribute to, and form the framework/rationale for, the aims and objectives of the present study.

Fig 1 Needs panel letters and I would find it easier to interpret if both had a similar orientation. Precise sample locations are needed. Are the locations of the three cores and the two deep samples noted here?

Fig. 2 Y axis states 'h' but caption states 'depth'. 'h' is undefined, but appears to be

height above bed. I know this depth scale is determined by translating the surface transect but it needs formal presentation and geometrical-dynamic argument and an error analysis. What flow model was used? If a flow model was not used, then at the very least the 3D surface geometry of the sample profile needs to be presented and the geometrical translation illustrated – in the main text. Is the glacier 100 m deep at present? Where in the glacier does this model relate to?

Fig. 3 Show uncertainly in elemental ratios. Panels need labelling.

Recommendations

I would combine all that is currently published relating to the chronology of MPG into the Introduction and then recast the aims to address a clear knowledge gap. For example, it is already known that the glacier has lost ∼40 m of ice since 1980. If so, roughly how many years of accumulation does this cover and, if we are not sure, then can that – along with the existing chronology - form the basis of rationale for a chronological study based on flow-line surface sampling. Why here and not in the upper glacier?

An age-depth model derived on the basis of the analysis of samples from the ablation area of an ablating glacier is not a trivial glaciological advance. I think this should form the main aim of the paper and be presented and argued in a logical and formal way, with relevant data presented in the main text and not supplementary information. Having done this, I would like to see a rigorous assessment and inclusion of all uncertainties involved in the age and depth scales, included in Figures such as current Fig. 2. I realise this cannot be achieved with great confidence, but I imagine it can be approximated.

(i) Methods, (ii) Results, and (iii) Interpretation/Discussion/Conclusions need to be separated clearly. As a minimum, Results need to be separated from Interpretation and Discussion.

---

## Referee Comment (RC2) · Wilfried Haeberli (Referee) · 3 Jul 2020

General

The authors provide results from a focused local study on remains of a glacier at Monte Perdido (MPG) in the Pyrenees. Their comprehensive analyses concern a question of quite fundamental relevance: are glaciers and, hence, the climate system now changing beyond natural, pre-industrial variability ranges? The main results of the analyses are that the maximum age of MPG can be constrained to the Roman period, and that no ice dating to the Little Ice Age remains present today. The results are interesting and certainly merit publication. They especially have the potential to encourage similar

studies in other regions of the world. Some parts need clarification and more precise presentation as explained below.

Sampling and age structure

Since the inferred age structure is central to the manuscript's main conclusions, it deserves a more clear and detailed presentation. The description of the ice sampling (e.g. on lines 151-163) is difficult to follow and should be clarified. The samples are obviously taken perpendicular to the stratigraphy along a profile at the surface of the stagnant, regularly layered ice patch. The assumption that this ice is cold and frozen to its bed may be reasonable, because this ice cannot warm up above 0°C in summertime but cool down far below 0°C in winter. This effect can explain the low flow velocities but not the ice stratigraphy, which must have been influenced by the active flow of the much larger glacier during the past millennia in question. This leads us to the following concrete questions that should be addressed in more detail:

(1) What exactly is the reasoning behind the inferred age structure of the remaining ice patch? Is it purely empirical from the dating or is it based on considerations of ice flow? The 14C dates are clustered in three different age groups, but the use of a linear interpolation needs better justification. In particular, why can the existence of further (presumably shorter) periods of hiatus or ice loss really be excluded from the presented evidence? Relatedly, if the only support for the hypothesis of a hiatus at 73 m is coming from a distinct dark layer (lines 233 ff., 302-303) – how does this interpretation of concentrated, impurity-rich dark layers fit with what is observed at the glacier surface today?

(2) How are distances along the surface profile transformed into values of (ice?) depths? Does "depth" relate to a former, thicker and less inclined ice body and, if yes, to which geometry/time exactly?

(3) What are "stratigraphic thicknesses" and how are they determined?
(4) If the ice is frozen to bedrock and stagnant, why did the authors find no evidence of neoglacial ice at the base and which process would have led to its removal?

(5) It also needs to be made more clear which part of the glacier was sampled (the lower portion?) and why the other (the upper portion?) was disregarded. Figure S2 should be included in the main text and supplemented with a zoom-in to the visual stratigraphy around the sampling sites for better visibility of the layering. Figure 4 suggests that neoglacial ice in the upper portion did not survive the Roman period, which is not supported by evidence in the manuscript. Are the authors assuming that this ice was removed by basal melting when the larger glacier was still warm-based, by thinning or ice flow? How does this align with the evidence for ice being frozen to bedrock now?

Radiometric and glacio-chemical analyses

The allocation of the samples to a position needs to be revised, at present much is left unclear to the reader. There seem to be two coordinates to consider: First, the position of the sampling site along the transect (MP1-100). Second, the depth below the surface / distance from bedrock. This information should be included in Table 1 to replace "sample depth (m from base)" – which is presumably referring to the distance from the glacier terminus? Again, a clear hypothesis should be stated why a systematic gradient in age of the samples in relation to their position on the glacier is expected? If the ice is stagnant, why is older ice expected closer to the terminus? The depth information should also be provided for the glacio-chemical datasets (especially Pb/Al and Hg of Figure 3).

The selection of 14C data for dating needs clarification, especially because a substantial number of samples is disregarded. The WIOC technique is state-of-the-art but only one WIOC sample is used to construct the chronology. Known difficulties with the interpretation of dating derived from macroscopic 14C, such as reservoir effects need to be addressed in more detail. Dark and dust-rich layers can be biased either through

incorporation of already "old" carbon (e.g. Saharan dust) or accumulate at the surface over a longer time period without ice formation. Regarding the pollen dating, which are presumably too old, the authors hypothesize that they originate from older ice which had melted and percolated through the ice. If this is true, how can such a process be excluded for the other radiocarbon dates?

Percolation of meltwater can also lead to redistribution of chemical impurities – would this be relevant at MPG and if not, why not? Along the same lines, it is important to give more attention to the glaciological settings of the site when interpreting the glacio-chemical records. Based on the presented hypothesis (Fig. 4), the MPG would have undergone substantial changes regarding its ice formation, possibly from a typical firnification process during cold periods to hiatus and melting during warm periods. An exposed glacier surface can lead to concentrated values of impurities, which would be more frequently the case in warm periods such as the roman or medieval period. In this sense, it is not clear that the heavy metals and their ratios should directly reflect any regional mining or smelting activities – this should either be removed or supplemented significantly by further discussion and justification. Notably, the connection between mining activities and heavy metal ice core records in the Alps was made at very high elevation locations (>4000 m asl) with a quasi-continuous snow sampling behavior.

Considering these points, the respective part of the manuscript dealing with the interpretation of the glacio-chemical analyses needs to be substantially revised and shortened. The main support for the conclusions of the manuscript provided by the impurity analysis is the absence of ice dating to the industrial period. This point has value for the manuscript. The relation to mining activities and chronological support through the comparison with the Marboré Lake record seems, at present, speculative.

Some minor technical comments can be found in the annotated file.

Wilfried Haeberli and Pascal Bohleber, 3 July 2020

Please also note the supplement to this comment:
https://tc.copernicus.org/preprints/tc-2020-107/tc-2020-107-RC2-supplement.pdf

[Figure]

**Supplement:**

- 1 The case of a southern European glacier disappearing
- 2 under recent warming that survived Roman and Medieval
- 3 warm periods
- 4 Ana Moreno1, Miguel Bartolomé2, Juan Ignacio López-Moreno1, Jorge Pey1,3, Pablo
- 5 Corella4, Jordi García-Orellana5,6, Carlos Sancho+, María Leunda7, Graciela Gil-
- 6 Romera8,1, Penélope González-Sampériz1, Carlos Pérez-Mejías9, Francisco Navarro10,
- 7 Jaime Otero-García10, Javier Lapazaran10, Esteban Alonso-González1, Cristina Cid11,
- 8 Jerónimo López-Martínez12, Belén Oliva-Urcia12, Sérgio Henrique Faria13,14, María José
- 9 Sierra15, Rocío Millán15, Xavier Querol16, Andrés Alastuey16 and José M. García-Ruíz1

[revised manuscript text omitted]

---

## Author Comment (AC1) · 8 Sep 2020

Answers to tc-2020-107 RC1 "The case of a southern European glacier disappearing under recent warming that survived Roman and Medieval warm periods".

Note: Reviewer 1 comments start with RC1 while author responses start with AR.

RC1. General comments: The paper is interesting and reports worthwhile chronological and elemental data, interpreted in terms of recent intense ablation removing 600 years of ice from MPG (despite the glacier surviving warm times before then). Unfortunately, as presented, I am not convinced the data analysis supports the main

conclusions and I cannot recommend the manuscript be published in its current form.

AR.We appreciate the interest on our manuscript and on the data presented. We provide our answers here to the three main points raised by Reviewer 1 hoping to solve his/her main concerns. Importantly, we think we can approach the chronological issue in a new revised version following his/her advice regarding (1) the translation from lateral surface samples to depth, (2) explanations about the samples that we discarded from the chronology, (3) detailed interpretation of the "debris-rich" layers and (4) relation among the 100 studied samples and the three recovered ice cores. Ideas about the order and organization of the main text are easy to include in a new version of the manuscript and certainly will help to improve its readability.

RC1.First and most importantly, I believe the chronology needs to be addressed with greater structure and formal rigour. For example: - Since this is so central to the paper's message, I find the translation from lateral surface samples to depth too difficult to follow in detail.

AR.We agree with this appreciation. Figure S2 showed a basic scheme of how we sampled the glacier ice and how we translate the surface samples to a depth profile. This figure was probably insufficient and not clear enough to understand our method. We provide an improved version of that figure (Figure 1, see below), which will be included in the main text of the revised manuscript as Figure 2. In that figure the bedding of the glacier ice is included with a downward slope of about 20 degrees. Although such a value has not been derived from local ice-thickness measurements, they are consistent with those derived from neighbouring GPR profiles further to the east (López-Moreno et al., 2019). This figure will qualitatively help the readers to better understand the problem under study and how the sampling was done. A more complete information on this topic will be included in the text of the revised version.

RC1. It appears seven of 17 age samples were dismissed from the analysis; these need detailed comment on each.
AR. This information was already included in lines 309-327 of main text, but at a location within the paper that was not suitable. We greatly value the suggestions by the reviewer regarding restructuring of the manuscript contents, and will follow his/her suggestions. Of course providing the reasons to discard 7 out of 16 dates is one of the most important aspects of our discussion on chronology, so in the revised version we will elaborate on the reasons to discard every sample.

Basically, we keep 8 samples from the first set of 9 bulk samples we dated. Those samples were the ones with more amount of organic material and we expected to get good 14C dates from them. The only one from that set we had to discard was MP46 that was out of order (younger than expected) and attributed to the presence of small pieces of plastic (too small to be removed) contaminating the sample. After obtaining the first 9 dates, we selected some other intervals where the dating information was insufficient but, unfortunately, the samples were not so good in terms of quantity and quality of the organic material. Then, two samples dated by WIOC technique provided too large errors due to the low amount of carbon (still, we included one of them in the age model); three samples from pollen concentrates were too old (several millennia older than the others) likely due to associated problems to concentrate pollen (Kilian et al., 2002) and, finally, two samples from material contained in filters where the mixing among carbonate debris, dust and variable organic matter provided incongruent results. In summary, we constructed the age model from 8 samples from the first set of 9 bulk samples and 1 from WIOC. The other ones were easy to discard following the explained issues.

RC1. The interpretation of debris-rich englacial layers as periods of ablation (concentrating the debris) needs a far more rigorous argument based on physical analysis and exclusion of alternative possibilities. At present, the reader does not know whether these are isochronous, or whether they deform passively or cut-across primary layering/stratification. Could they be basally-derived? How are supply-rate variations excluded?

AR. Although alternative explanations cannot be completely excluded, we have firm reasons to believe that the sequence of debris-rich layers observed along the sampled profile correspond to the primary stratification of debris deposited at the surface of the glacier and are therefore isochronous layers (except for the cases in which the primary layers became merged due e.g. to intense melting episodes and/or low surface accumulation periods). Note, first, that the distribution of layers is rather regular and extends laterally as shown in Fig. 1 and in former Fig. S2, as would be expected for a stratification stemming from the original deposition at surface of snow and debris. Also note that the isochronal layers in a glacier emerge in the ablation area, but this is consistent with our case study, as our sampled profile corresponds entirely to the ablation zone. The reality is more complex, as our glacier has been shrinking and retreating since the end of the LIA, but the situation would be approximately as depicted below (Figure 2).

The reviewer questions whether the observed debris layers could be basally-derived. We believe that they are not. The main reason is that, in general, debris layers transverse to the glacier flow direction correspond to thrust faults developed near the glacier margin due to the compressional stress regime, most often associated to the transition, when we approach the glacier terminus, from warm-based to cold-based ice near the terminus. The warm-based ice slides over its bed, while the cold-based one does not, so strong compressional stresses develop, which cannot be accommodated by creep and thrusts develop. These thrust faults may reach the surface or terminate englacially (blind thrusts; see figure GM-a below). Basal debris is incorporated into these thrusts, sometimes reaching the surface. When they do so, due to surface melting, they often produce large accumulations of debris (often in the form of pinacles or pinnacle ridges along the debris layer; see figs. GM-a and BG below); intense melting episodes can spread this debris over the glacier surface. We believe that the debris layers observed at the surface of MPG are not subglacially-derived, because the hypothetical thrust faults supplying subglacial till to the glacier surface would be limited to the terminal zone, while in MPG the debris layers are observed all-along the sampled profile covering the entire lower glacier. Neither the mentioned large accumulations of debris, nor pinacles, are observed in MPG. Furthermore, if such thrust faults close to the ice margin would form in an advancing glacier (such as MPG during the LIA), they would form progressively in front of each other as the ice margin advances. Upon retreat, they would appear as a moraine-mound complex as shown in figures BG and GM-b below. However, in the proglacial zone of MPG, the only hummocky moraines have been identified as push moraines created by glacier advance before 1850 CE (Serrano and Martín-Moreno, 2018).

Fig. 3: Fig. 7 of Graham and Midgley (2000).

Fig. 4: Fig 9.6 of Bennet and Glasser (1996) – modified from Fig. 2.36 of Hambrey (1994).

Finally, we note that, as indicated in lines 283-287 of the original manuscript, "The debris layers were composed of detrital, silty-sandy size deposits, likely coming from wind-blown particles (e.g. black carbon-rich particles, dust) and from erosive processes of the limestone catchment, including the fall of gravel-sized particles from the surrounding cliffs" and they show no evidences of subglacially-derived glacier till.

Regarding the last concern of the reviewer, on how are supply-rate variations excluded, we note that our hypothesis does not necessarily exclude variations of debris supply-rate. In fact, higher supply rates are expected during warmer periods, in which more exposed rock surface should be available to supply falling debris from adjacent slopes and a larger extent of ice/snow-free terrain would supply a higher amount of wind-blown dust. This is not inconsistent at all with our comment in lines 288-291 of the original manuscript that "the frequency of debris layers increases towards the top of the glacier, where these layers are most abundant. We consider the accumulation of debris layers to be indicative of reduced ice accumulation and dominance of ablation periods."

RC1. How were the three core samples (and two deep samples) combined with theâĹij100 surface samples?

AR. The cores and the two samples from the front belong to a different area of the glacier, more dynamic, and for this reason they have not been combined with the 100 surface samples analysed in the present study. We recognise that, in the first submitted version of the manuscript, our writing could suggest that there was a relation among the cores and the 100 samples. Thus, in the new version, we will focus on explaining the sampling on the slope as the method selected to sample the glacier ice, and will justify the lack of coring by noting that glacier conditions did not allow obtaining a complete ice sequence by coring. Text reads as this: Ice sampling in MPG was carried out in September 2017 along an ordered chrono-stratigraphical sequence covering from the oldest to the newest ice preserved in the glacier (Fig. 2). Unfortunately, coring was not possible since none of the glacio-meteorological and topographical criteria required to obtain a preserved ice-core stratigraphy, such as low temperatures to prevent water percolation, or a large extension and flat surface topography to minimize the influence of glacier flow (Garzonio et al., 2018), are currently met in the glacier.

RC1. Since the annual 0 deg. C contour is at 3,000 m why wasn't snow/firn/ice sampled from the upper glacier. This glacier seems all to lie above this elevation and may therefore be accumulating, providing an undisturbed record of accumulation change?

AR. We could not reach the upper glacier in a safe way for sampling. Thus, up to now, there are no available samples, neither any age information from the upper glacier.

RC1. For me, too much key material seems to be given in the supplementary information rather than forming a central part of the argument of the main paper.

AR. We agree with this appreciation and have included all the supplementary information in the main paper.

RC1. For comparison, what elemental values have been measured in recent ice (not snow/firn)?

AR. Unfortunately, we cannot compare elemental values from current ice in Monte Per-
dido glacier since ice is not being formed at present, as the glacier has virtually no accumulation zone (the accumulation-area ratio tends to zero). We obtained the same information derived from the lack of 210Pb on the surface ice samples. Neither we have analyses of MPG snow. However, we have several measurements of snow from other zones in the Pyrenees at similar altitude, and found that the variability in elementary values from one site to another, or even from a sampling campaign to another, is enormous (see recently published paper Pey et al. 2020). It would be incorrect to use such values to compare with fossil ice in MPG. On the contrary, sampling atmospheric aerosols in Ordesa National Park station appears better suited to compare with MPG, since we can average over two years to get a more comparable value. Doing this, several elements appear clearly enriched in present-day atmosphere compared with MPG samples.

RC1. Second, substantial relevant material relating to the chronology of MPG appears later in the manuscript when I believe it should be summarised in full in the Introduction, directing the specific aim and objectives of this study. In my view, it's fine to introduce related information in the Discussion (such as the chronology of lakes in the area or of broader Alpine glaciers), but material directly relating to MPG - and particularly the focus of this submission (i.e., its chronology) - should form relevant background material in the Introduction.

AR. The only materials related to the chronology of MPG are the few dates corresponding to the moraines. We include them now in the Introduction as suggested by Rev. 1 and use them to state more clearly our aims.

RC1. Third, I found several of the sections and their contents to be confusing. For example, what I consider to be results are presented in the Field Site and Methods section, and I would separate Results from Discussion.

AR. Following this suggestion, we have separated Results and Discussion and included some dating information that was previously in Methods as Results. We hope that this

new organization of the contents, together with the inclusion of all of the material from the Suppl. Materials in the main text, makes this paper more clear and readable.

RC1. Specific comments Note, the manuscript includes many slight grammatical and typographical errors, as a few stylistic imperfections, which I have mostly not corrected. For me, the meaning was almost always clear, if not grammatically perfect.

AR.We have carefully reviewed the text looking for grammatical and typographical errors and corrected them.

RC1. Line/Location Comment/Suggestion 1 Title is awkward because 'that' is unspecified.

AR. We have changed the title to "The case of a southern European glacier that survived Roman and Medieval warm periods but is disappearing under recent warming". Thus, we better specify that is the glacier the one disappearing today but surviving during other warm periods.

RC1. 47 I'd avoid 'proves' unless the sentence states 'if we assume the chronological model is correct...'.

AR. We do not think that we need to assume that the chronological model is correct... it is based on 14C dates and the uncertainty is also considered. Yet, we can change the word "proves", which is probably too strong, to "evidences".

RC1. 54-55 This claim needs to be supported by a rigorous analysis. AR. We claim that the current warming is unprecedented in the context of last 2 kyr because the loss of ice from MPG in the last decades has been the greatest since we have records. The last sentence was probably not properly placed in the abstract, as it is based on conclusions from previous studies ("We demonstrate that we are facing an unprecedented retreat of the Pyrenean glaciers whose survival is compromised beyond a few decades") and thus we have removed it.

RC1. 94-105 I'd invert this: state the aim of the paper and then state how it was

achieved. AR. We think the reviewer was confused by the wrong use of tense of verbs in this paragraph. Now, those explaining previous studies have been changed to past tense, and we state the aim of this paper, and how it was achieved, as suggested by Rev. 1.

RC1. 141 Data need to be presented and analysed to substantiate the claim that '(GPR and modelling)...suggested that the oldest ice could be located in these areas.' This claim is central to the chronology presented in the manuscript.

AR. That section has been removed, since it referred to the three cores which did not cover the whole sequence and thus were not studied.

RC1. 161 1 m of what? A Jacob's staff should be described, or just use 'staff'.

AR.Ok, done

RC1. 175 Which 'uppermost five samples'?

AR. We selected 5 samples towards the top of the sequence to have the most recent ones. In the text we now indicate which are those samples.

RC1.175-6 These are Results

AR. Ok, moved to results

RC1.176-8 This is Interpretation

AR. Ok, moved to discussion

RC1. 186-96 These are a mixture of Results and Interpretation. What values would be expected or have been measured on accumulating glaciers in similar settings?

AR. We moved some sentences to Results and some to Discussion.

RC1. 209 How and where did these samples come from? How were they treated?

AR. They are just two samples in the sequence of 100 samples. They correspond to

samples 67 and 81 and were selected because in that portion of the record the dating obtained by the first 9 samples of bulk material was insufficient. We thought those samples may have more organic material and were easy to date. ... The pre-treatment is already indicated in the text and the treatment in the 14C lab was similar to any other sample just removing the filter at the first stage.

RC1. 231-2 Why a second-order polynomial/quadratic? If there is a theoretical basis for such a relationship, then that should be presented. If there is not, then what justification is there for this form?

AR. Before explaining the reasoning behind this approach, we note that the model was not built using a second-degree polynomial, and thanks to the referee we have spotted this error. We initially took this model as a quality control for the 7 dates amongst the 16 from which we presumed low quality due to various reasons (discussed in section 4.1 and along this letter). Using a second-degree polynomial approximation, instead of spline or other interpolations, higher degree polynomials, or a Bayesian approach, has been evidenced to be the best approach when there is a low proportion of dates in comparison with the potential changes in sedimentation rate (Telford et al., 2004). In such a way we still account for potential non-linear accumulation, though it rarely happens in nature, while we reduce the number of assumptions when increasing the regression into higher orders. We insist that this approach was used just to double test the quality of the samples that we were discarding.

Our final model was then made with the 9 remaining dates, where we set a hiatus at sample D-AMS 025298 (uncalibrated 14C 1011$\pm$ 25, 2700 cm, equivalent to 73 m depth following a bottom-up sampling strategy, as explained in Figure 2 above). We then run a linear regression that, given our scarce prior knowledge on the record sedimentation and according to the particular nature of our archive, seemed the most reasonable one, reducing the overfitting to noise of splines or higher-order regressions, still not forcing the model though all dates as a piecewise linear interpolation would do.

[Figure]

RC1. 234-8 This argument relating to debris concentration by ablation/low accumulation needs substantial background and argument, including arguments for this interpretation and against alternative interpretations of englacial debris bands. For example, how are supply-rate variations or a source at the glacier bed excluded?

AR. We are referring to the primary ice stratigraphy, which is evidenced by the alternance of layers with more debris (darker layers) and debris-poor cleaner ice (lighter in colour). This alternance occurs at a metric scale and appears all along the ice sequence (Figure 1 above). We interpret that these layers result from periods (summer?) with less ice formation and more "debris" supply. Interestingly, we observe this alternance more clearly between 67 and 72 "meters", where dark layers are thicker and are more closely spaced. From that interval we dated several samples (MP-67, MP-68, MP-69, MP-70, MP-73), obtaining dissimilar results, not in order, and covering a time interval from 600 to 1200 CE. Our hypothesis is that ice was formed at a very low rate during that period (600-1200 CE). Or, even if it was formed at a similar rate as in other periods, at some point it melted. Once most of the ice is melted, the accumulation of the debris particles becomes more evident. Here, the "debris" does not seem to correspond to material eroded at the glacier bed and entrained in the basal ice or transported to the glacier surface through thrust faults near thee terminus, but just to material mostly transported by winds and deposited on the ice surface. Further comments on the debris layers have been included in the answer to the general comments.

RC1. 288-91 These arguments need formalising, expanding and presenting as a logical progression of argument; currently, it is difficult to evaluate the accuracy of the chronology because this logical progression in argument (supported by illustrated data) is missing or split up through the paper and supplementary information.

AR. We agree with this comment about the lack of a logical progression in our argumentation, since the discussion on chronology was indeed split up in various sections of the manuscript. This is now corrected and, after an explanation of the methods

employed, all the discussion concerning the age model is included as Results in the revised version. We have substantially changed the structure and organization of the manuscript, and are confident in that the revised version has improved its readability.

RC1. 281- I find this section difficult to follow since in some places Results have already been presented and they are only referred to here (e.g., 306-8) or Results are mixed with Interpretation, and in some case Methods are included here (318-27). In relating to the last point, I'd discuss sample removal from analysis in Methods and not Results.

AR. Yes, we agree and we have worked on that direction in the new version.

RC1. 328-30 These criteria should be discussed in full and presented in Methods rather than Results.

AR. Yes, we agree and have worked in that direction in the new version.

RC1. 333 I'd bullet or number these three main periods and interpret consistent/repeated data across all three.

AR. Ok, we have indicated the three periods by numbers.

RC1. 361- I'd move much of this into a dedicated Discussion section

AR. We have moved part of this into the Results section (explanation of trace elements values in present-day aerosols and in the glacier ice), and partly into a new section in the Discussion (comparison with another paleoclimate record nearby, the Marboré lake).

RC1. 412-26, 434-7 & 467-70 I'd move this published material relating specifically to the chronology of MPG to the Introduction. That way it would contribute to, and form the framework/rationale for, the aims and objectives of the present study.

AR. Done. That information in the introduction helps to state the objectives of the present study.

RC1. Fig 1 Needs panel letters and I would find it easier to interpret if both had a similar orientation. Precise sample locations are needed. Are the locations of the three cores and the two deep samples noted here?

AR. Panel letters are included and map changed to have the same orientation as the picture. The three cores are not indicated, since we have not studied them, so just the transect with the 100 samples is shown.

RC1. Fig. 2 Y axis states 'h' but caption states 'depth'. 'h' is undefined, but appears to be height above bed. I know this depth scale is determined by translating the surface transect but it needs formal presentation and geometrical-dynamic argument and an error analysis. What flow model was used? If a flow model was not used, then at the very least the 3D surface geometry of the sample profile needs to be presented and the geometrical translation illustrated – in the main text. Is the glacier 100 m deep at present? Where in the glacier does this model relate to?

AR. All these questions posed by Rev1 indicate that the way we translated sample number to depth scale was not clear at all. As indicated in the main point of this letter, more text is added to explain this issue and Fig S2 is now included in the main text and improved with more information about glacier ice bedding (See Figure 1 above).

RC1. Fig. 3 Show uncertainly in elemental ratios. Panels need labelling.

AR. Panels are labelled. Uncertainty is indicated in methods, we do not think that it is necessary to include it in the figures for every sample.

RC1. Recommendations I would combine all that is currently published relating to the chronology of MPG into the Introduction and then recast the aims to address a clear knowledge gap. For example, it is already known that the glacier has lost 40 m of ice since 1980. If so, roughly how many years of accumulation does this cover and, if we are not sure, then can that –along with the existing chronology - form the basis of rationale for a chronological study based on flow-line surface sampling. Why

here and not in the upper glacier? An age-depth model derived on the basis of the analysis of samples from the ablation area of an ablating glacier is not a trivial glaciological advance. I think this should form the main aim of the paper and be presented and argued in a logical and formal way, with relevant data presented in the main text and not supplementary information. Having done this, I would like to see a rigorous assessment and inclusion of all un-certainties involved in the age and depth scales, included in Figures such as current Fig. 2. I realise this cannot be achieved with great confidence, but I imagine it can be approximated.(i) Methods, (ii) Results, and (iii) Interpretation/Discussion/Conclusions need to be separated clearly. As a minimum, Results need to be separated from Interpretation and Discussion.

AR. We really appreciate this final summary from Rev1 and are sure that we can follow his/her advice in a new revised version of the manuscript. The most problematic point regarding age-depth relationships is improved by including Fig S2 in the main text (in fact, all information in the Supp. Materials is now in the main text) and by more detailed explanations. However, we cannot present a rigorous assessment of uncertainties in depth scale since we do not know the inclination angle of the glacier ice layers to translate our sample numbers to real depth. We can approximately calculate the bedding tilt to finally have 30 m of ice sequence as happens in the easternmost section of the glacier. But still it would be unprecise and speculative. Therefore, we do not use the term "depth" anymore in the age model construction but "sample number" or "sample ID", which is more accurate. Regarding age uncertainties, they are included in the age-depth model. The organization separating methods / results / discussion really helps to understand the ideas and outcomes of this study. Similarly, including in the introduction more information about what we know about this glacier helps to better state our aims.

AR. References cited:

Bennet, M.R., Glasser, N.F., 1996. Glacial Geology. Ice Seets and Landforms. John Wiley and Sons, Chichester.

[Figure]

Hambrey, M. , 1994. Glacial Environments. UCL Press, London.

Graham, D.J., Midgley, N.G., 2000. Morain-formation by englacial thrusting: the Younger Dryas moraines of Cwm Idwal, Noth Wales. In: Maltman, A.J., Hubbard, B., Hambrey, M.J. (Eds.). Deformation of Glacial Materials. Geological Society Seccial Publlication No. 176, p. 321-336. Geological Sciety, London.

Kilian, M.R., van der Plicht, J., van Geel, B., Goslar, T., 2002. Problematic 14C-AMS dates of pollen concentrates from Lake Gosciaz (Poland). Quaternary International 88, 21–26. https://doi.org/10.1016/S1040-6182(01)00070-2

López-Moreno, J.I., Alonso-González, E., Monserrat, O., Del Río, L.M., Otero, J., Lapazaran, J., Luzi, G., Dematteis, N., Serreta, A., Rico, I., Serrano-Cañadas, E., Bartolomé, M., Moreno, A., Buisan, S., Revuelto, J., 2019. Ground-based remote-sensing techniques for diagnosis of the current state and recent evolution of the Monte Perdido Glacier, Spanish Pyrenees. J. Glaciol. 65, 85–100. https://doi.org/10.1017/jog.2018.96

Pey, J., Revuelto, J., Moreno, N., Alonso-González, E., Bartolomé, M., Reyes, J., Gascoin, S., López-Moreno, J.I., 2020. Snow Impurities in the Central Pyrenees: From Their Geochemical and Mineralogical Composition towards Their Impacts on Snow Albedo. Atmosphere 11, 937. https://doi.org/10.3390/atmos11090937

Telford, R.J., Heegaard, E., Birks, H.J.B., 2004. All age-depth models are wrong: but how badly? Quaternary Science Reviews 23, 1–5. https://doi.org/DOI:10.1016/j.quascirev.2003.11.003

[Figure]

[Figure]

**Fig. 1.** This would be the new Figure 2 in the revised version, with an image of the glacier surface where we conducted the sampling and a scheme with the position of the 100 samples taken along the slope.

**Fig. 2.** In (1) we see the particle trajectories (blue) and the associated isochrones (red), emerging below the ELA; in (2) the trajectories have been removed to show only the isochrones; (3) shows subsequent

a

Debris-rich thrusts

glacier
ice

Décollement
surface

Bedrock

b

moraine-mound complex

Imbricately stacked mounds

glacier
ice

**Fig. 3.** Fig. 7 of Graham and Midgley (2000).

[Figure]

**Fig. 4.** Fig 9.6 of Bennet and Glasser (1996) – modified from Fig. 2.36 of Hambrey (1994).

---

## Author Comment (AC2) · 8 Sep 2020

Answers to tc-2020-107 RC1 "The case of a southern European glacier disappearing under recent warming that survived Roman and Medieval warm periods"

Note: The referee comments start with "RC2" and the author's answers with "AR".

RC2. General The authors provide results from a focused local study on remains of a glacier at Monte Perdido (MPG) in the Pyrenees. Their comprehensive analyses concern a question of quite fundamental relevance: are glaciers and, hence, the climate system now changing beyond natural, pre-industrial variability ranges? The main

results of the analyses are that the maximum age of MPG can be constrained to the Roman period, and that no ice dating to the Little Ice Age remains present today. The results are interesting and certainly merit publication. They especially have the potential to encourage similar studies in other regions of the world. Some parts need clarification and more precise presentation as explained below.

AR. We acknowledge the positive view of Drs. Haeberli and Bohleber regarding the importance of this study and the interest to publish it in the TC journal. We provide answers below to their main concerns about sampling and age structure and about radiometric and glacio-chemical analyses.

**RC2. Sampling and age structure**

RC2. Since the inferred age structure is central to the manuscript's main conclusions, it deserves a more clear and detailed presentation. The description of the ice sampling (e.g. on lines 151-163) is difficult to follow and should be clarified. The samples are obviously taken perpendicular to the stratigraphy along a profile at the surface of the stagnant, regularly layered ice patch. The assumption that this ice is cold and frozen to its bed may be reasonable, because this ice cannot warm up above 0C in summertime but cool down far below 0C in winter. This effect can explain the low flow velocities but not the ice stratigraphy, which must have been influenced by the active flow of the much larger glacier during the past millennia in question. This leads us to the following concrete questions that should be addressed in more detail:

(1) What exactly is the reasoning behind the inferred age structure of the remaining ice patch? Is it purely empirical from the dating or is it based on considerations of ice flow? The 14C dates are clustered in three different age groups, but the use of a linear interpolation needs better justification. In particular, why can the existence of further (presumably shorter) periods of hiatus or ice loss really be excluded from the presented evidence? Relatedly, if the only support for the hypothesis of a hiatus at 73 m is coming from a distinct dark layer (lines 233 ff., 302-303) – how does this
interpretation of concentrated, impurity-rich dark layers fit with what is observed at the glacier surface today?

AR. We agree about the problems posed by the unknown ice stratigraphy, which certainly was the result of the glacier evolution during the last millennia. This problem is guite common to many high-mountain glaciers, even when coring is possible, since their detailed inner structure by physical analyses such as GPR is rarely available. When we applied ice flow models to MPG (using Elmer-ICE, in particular) the results indicated that the ice in MPG was at most 200 years old (the longest possible travel time of an ice particle deposited at the upper part of the current lower glacier), which was in contradiction with the obtained ages (210Pb, 137Cs, 14C). However, these models were just applied -given the lack of input data- assuming a steady-state surface (that of the current glacier), which is clearly incorrect. One explanation contributing to explain the old ages found is that the ice has been for long periods frozen to bedrock, and hence nearly stagnant. Moreover, we can just speculate about the inclination of the glacier bed necessary to account for the 30 m of maximum ice thickness previously measured by GPR in other sector of the glacier (López-Moreno et al., 2019). Our hypothesis that the oldest ice is found towards the base of the sequence and the newest towards the top is based on general arguments on ice flow of mountain glaciers. 14C dates confirmed this hypothesis, but we agree with Reviewers in that we cannot totally exclude the presence of other shorter hiatus besides the one at 67-73 m, which was the most evident. We set the hiatus at 73 m to be able to construct an age-depth model and use different interpolations below and above (before and after) it, depending on the growth rate indicated by the dating results.

RC2. (2) How are distances along the surface profile transformed into values of (ice?) depths? Does "depth" relate to a former, thicker and less inclined ice body and, if yes, to which geometry/time exactly?

AR. The reviewer is right about the "depth" concept here. It is related to our consideration of horizontal ice layers, since we obtained one sample every meter, measur-

**TCD**
ing these meters with the Jacob staff without taking any inclination into consideration. Therefore, the 100 samples would represent 100 m of thickness in the ice sequence if the ice bedding were horizontal. The exact thickness is unknown, but in the order of 30 m in the easternmost sector of the glacier. So, the internal layers of the MPG must be very inclined. An improved version of Figure S2 may help to understand this problem (Fig. 1).

RC2. (3) What are "stratigraphic thicknesses" and how are they determined?

AR. We are not using stratigraphic thickness anymore since we cannot measure it, as indicated in our response just above. We have now included "Sample ID" instead of "Depth" in the age-model figure.

RC2. (4) If the ice is frozen to bedrock and stagnant, why did the authors find no evidence of neoglacial ice at the base and which process would have led to its removal?

AR. We have constructed the figures about the evolution of the glacier based on what we have found sampling and dating the ice. Then, since we have not found the Neoglacial ice it likely was melted away at some point between 5000 years BP and the Roman Period. It is also possible that some ice from the Neoglacial periods remains in the base of the glacier but it was not cut by the vertical section of our surface sampling profile (see figure above), so we did not sample it. We have included this possibility in the new version of Figure 4 (see below Fig. 2).

RC2. (5) It also needs to be made more clear which part of the glacier was sampled (the lower portion?) and why the other (the upper portion?) was disregarded. Figure S2 should be included in the main text and supplemented with a zoom-in to the visual stratigraphy around the sampling sites for better visibility of the layering. Figure 4 suggests that neoglacial ice in the upper portion did not survive the Roman period, which is not supported by evidence in the manuscript. Are the authors assuming that this ice was removed by basal melting when the larger glacier was still warm-based, by thinning or ice flow? How does this align with the evidence for ice being frozen to

TCD
bedrock now?

AR. The upper portion of the glacier was disregarded because it is really small today, with a slope very steep and the access for sampling was dangerous. Unfortunately, we do not have any date from that sector. Fig. S2 is now improved and will be included in the revised version of the manuscript (Fig. 1) with more details about layering and a closer picture of the glacier.

Regarding the Neoglacial ice survival, we can just hypothesize that it melted away at some point before or during the Roman period, since we don't find it in our dated samples. But, of course, it can remain in the upper glacier. It is also true that it may be found below the Roman ice that we sampled and is not exposed at the glacier surface today. We have to consider all of these possibilities in the new revised version of the manuscript and update Figure 4 with them. See below a first version of Figure 4 keeping part of the Neoglacial ice during the 2000-year period (Fig. 2).

We do not know when the glacier was frozen to bedrock. That situation might have happened once the glacier became sufficiently thin (after the Roman period? After the MCA?) and probably did not become warm-based anymore.

RC2. Radiometric and glacio-chemical analyses

RC2. The allocation of the samples to a position needs to be revised, at present much is left unclear to the reader. There seem to be two coordinates to consider: First, the position of the sampling site along the transect (MP1-100). Second, the depth below the surface / distance from bedrock. This information should be included in Table 1 to replace "sample depth (m from base)" – which is presumably referring to the distance from the glacier terminus? Again, a clear hypothesis should be stated why a systematic gradient in age of the samples in relation to their position on the glacier is expected? If the ice is stagnant, why is older ice expected closer to the terminus? The depth information should also be provided for the glacio-chemical datasets (especially Pb/Al and Hg of Figure 3).

TCD
AR. We totally agree with this comment about the difference among "position in the transect", "depth from the base" and "sample ID". The reviewer is right and we have replaced any reference to ice thickness or ice depth to sample position or sample ID. We expected the older ice closer to the terminus as a consequence of the bedding and due to the steep slope present today. Ice layers are cut by the glacier surface and older layers should appear closer to the terminus if they are tilted (see Figure 2 above). It is evident that more information on this is needed in the manuscript and will be included in the revised version. We do not have the depth information for Figure 3 but can include sample ID (from 0 to 100), if necessary.

RC2. The selection of 14C data for dating needs clarification, especially because a substantial number of samples is disregarded. The WIOC technique is state-of-the-art but only one WIOC sample is used to construct the chronology. Known difficulties with the interpretation of dating derived from macroscopic 14C, such as reservoir effects need to be addressed in more detail. Dark and dust-rich layers can be biased either through incorporation of already "old" carbon (e.g. Saharan dust) or accumulate at the surface over a longer time period without ice formation. Regarding the pollen dating, which are presumably too old, the authors hypothesize that they originate from older ice which had melted and percolated through the ice. If this is true, how can such a process be excluded for the other radiocarbon dates?

AR. Our first option for dating would have been the WIOC technique, but we had not enough material in most cases. We did not preserve all the ice samples frozen, just a few of them for studying bacteria and virus. Thus, we were able to attempt WIOC dating in just two samples that were frozen and were of larger size. Unfortunately, they had still very low amount of carbon for dating and the errors were too large. One had to be discarded because of that reason, and the other is included in the age model. More and better organized information is now included about the reasons to reject some of the samples. We agree about the possible reservoir effect when dating dark and dust-rich layers where organic and inorganic material is mixed but this unfortunately Interactive comment

is still difficult to avoid. For the future, Dissolved Organic Carbon (DOC) technique for dating may replace WIOC and provide more accurate results with less amount of sample (Fang et al., 2020).

However, we are not too sure about the reasons to obtain so old dates with pollen samples. Dating with pollen concentrates can be an accurate tool for chronological reconstruction, employed in many studies (González-Sampériz et al., 2006). However, obtaining old dates from pollen is a quite common problem not yet solved in the literature (Kilian et al., 2002). Another reason to exclude pollen samples for 14C consideration in MPG age model -not discussed in the manuscript- is that the obtained palynological spectra from the same samples that were dated is not coherent with well-known palynological records from the region. In the MPG area we have a detailed palinological study in Marboré lacustrine sequence which is totally different in taxa and abundance of that obtained from coetaneous MPG samples. Therefore, we suspect the pollen samples were contaminated somehow and we can not use them. Additionally, the information of pollen studies in active glaciers (specially related to the mechanisms of deposition and preservation) is scarce and thus not solves this problem. In any case, the three dated samples in MPG coming from pollen concentration are not consistent with the other onesand should be excluded.

RC2. Dark and dust-rich layers

RC2. Percolation of meltwater can also lead to redistribution of chemical impurities – would this be relevant at MPG and if not, why not? Along the same lines, it is important to give more attention to the glaciological settings of the site when interpreting the glacio-chemical records. Based on the presented hypothesis (Fig. 4), the MPG would have undergone substantial changes regarding its ice formation, possibly from a typical firnication process during cold periods to hiatus and melting during warm periods. An exposed glacier surface can lead to concentrated values of impurities, which would be more frequently the case in warm periods such as the roman or medieval period. In this sense, it is not clear that the heavy metals and their ratios should directly reflect any
regional mining or smelting activities – this should either be removed or supplemented significantly by further discussion and justification. Notably, the connection between mining activities and heavy metal ice core records in the Alps was made at very high elevation locations (>4000 m asl) with a quasi-continuous snow sampling behavior.

AR. We appreciate these comments and partially agree about the problems of percolation and redistribution of chemical impurities, which are relevant for many mountain glaciers in the current climate context. Still, we think that we can assume that most of the material is in place since it correlates well with the Marboré geochemical record, now published (Corella et al., 2021), and supports the chronology indicating the absence of ice from the industrial period. We agree that an exposed glacier surface can lead to concentrate impurity values, but these impurities would still have the same origin if they come from atmospheric aerosols, and can still be interpreted as a result from mining activities. Nevertheless, taking into account the problems of redistribution of chemical impurities due to percolation, we have to be more moderate with our interpretation of these elements. The text will be modified accordingly, presenting the interpretation of Pb/Al peak in the Roman times as Roman mining only as a possibility, and including the issue of percolation. We will also reflect in the text that the ice cores in the Alps where Pb/Al was considered as resulting from Roman-time mining activities were more continuous and located at higher altitudes, thus not strictly comparable to our site.

RC2. Considering these points, the respective part of the manuscript dealing with the interpretation of the glacio-chemical analyses needs to be substantially revised and shortened.

AR. Yes, we agree, as indicated above and modified the text accordingly.

RC2. The main support for the conclusions of the manuscript provided by the impurity analysis is the absence of ice dating to the industrial period. This point has value for the manuscript. The relation to mining activities and chronological support through the
comparison with the Marboré Lake record seems, at present, speculative.

AR. The comparison with Marboré Lake was carried out using an age scale obtained independently (age model from MPG presented in this study and age model from Marboré lake presented in Corella et al., 2021). Thus, the similarity of the Pb/AI records in both archives (lake and glacier) is at least interesting to show. We have given less weight to the interpretation of Pb/AI as mining in this new version, since we agree in that it can come from other sources. But we would like to keep the graph showing Marboré and MPG records together. From that graph, we will highlight the absence of ice dating to the industrial period, as suggested by the Reviewers.

RC2. Some minor technical comments can be found in the annotated file.

Wilfried Haeberli and Pascal Bohleber, 3 July 2020

AR. Many thanks for all these comments and suggestions to improve our manuscript.

AR. References cited

Corella, J.P., Sierra, M.J., Garralón, A., Millán, R., Rodríguez-Alonso, J., Mata, M.P., de Vera, A.V., Moreno, A., González-Sampériz, P., Duval, B., Amouroux, D., Vivez, P., Cuevas, C.A., Adame, J.A., Wilhelm, B., Saiz-Lopez, A., Valero-Garcés, B.L., 2021. Recent and historical pollution legacy in high altitude Lake Marboré (Central Pyrenees): A record of mining and smelting since pre-Roman times in the Iberian Peninsula. Science of The Total Environment 751, 141557. https://doi.org/10.1016/j.scitotenv.2020.141557

Fang, L., Jenk, T., Singer, T., Hou, S., Schwikowski, M., 2020. Radiocarbon dating of alpine ice cores with the dissolved organic carbon (DOC) fraction. The Cryosphere Discussions 1–26. https://doi.org/10.5194/tc-2020-234

González-Sampériz, P., Valero-Garcés, B.L., Moreno, A., Jalut, G., García-Ruiz, J.M., Martí-Bono, C., Delgado-Huertas, A., Navas, A., Otto, T., Dedoubat, J.J., 2006. Climate variability in the Spanish Pyrenees during the last 30,000 TCD
yr revealed by the El Portalet sequence. Quaternary Research 66, 38–52. https://doi.org/10.1016/j.yqres.2006.02.004

Kilian, M.R., van der Plicht, J., van Geel, B., Goslar, T., 2002. Problematic 14C-AMS dates of pollen concentrates from Lake Gosciaz (Poland). Quaternary International 88, 21–26. https://doi.org/10.1016/S1040-6182(01)00070-2

López-Moreno, J.I., Alonso-González, E., Monserrat, O., Del Río, L.M., Otero, J., Lapazaran, J., Luzi, G., Dematteis, N., Serreta, A., Rico, I., Serrano-Cañadas, E., Bartolomé, M., Moreno, A., Buisan, S., Revuelto, J., 2019. Ground-based remote-sensing techniques for diagnosis of the current state and recent evolution of the Monte Perdido Glacier, Spanish Pyrenees. J. Glaciol. 65, 85–100. https://doi.org/10.1017/jog.2018.96

**TCD**
**Fig. 1.** This would be the new Figure 2 in the revised version, with an image of the glacier surface where we conducted the sampling and a scheme with the position of the 100 samples taken along the slope.

---

## Author Response (AR1)

Zaragoza, 16th October 2020

Dear Dr. Farinotti,

I am submitting the revised version of our TC-2020-107 manuscript entitled "**The case of a southern European glacier that survived Roman and Medieval warm periods but is disappearing under recent warming**", co-authored by myself and colleagues, to be considered for publication in *The Cryosphere*.

In this new version we have followed closely the suggestions indicated by two previous reviewers. The main changes can be summarized as follows:

- Sampling methodology: our sampling procedure, including details about (1) the relation among the cores and the transect, (2) the translation of sample ID to thickness or (3) the order of the stratigraphical sequence (from old to young ice layers), were not clear in previous version (comments from RC1 and RC2) and are improved in this new one. In addition, new Figure 2 includes details of sampling and ice bedding.
- Chronology: we have included more clearly the criteria we followed to discard several 14C samples and more details on the methodology to select the samples for dating. How the age model was constructed is more detailed now, including the role of the dark debris-rich layers and their meaning as isochrones.
- Organization: we have followed RC1 advice about several changes in the manuscript structure such as including more information about MPG in the introduction, separating Results and Discussion and removing the supplementary material. Now all the information is in the main text, together with four tables and five figures.
- Trace elements: interpretation of Pb/Al ratio as the result of mining activities is reduced as suggested by RC2 and used the comparison to Marbore Lake record to highlight the lack of ice from the Industrial Era.

We hope this new version was suitable for *The Cryosphere*, and hope that it will fulfil your expectations. I will be happy to answer any question you might have regarding this study.

Yours sincerely,

annenero

Ana Moreno Caballud

Avda. Montañana, 1005 50.059Zaragoza (España) TEL.:976-369-393 Answers to tc-2020-107 RC1 "The case of a southern European glacier disappearing under recent warming that survived Roman and Medieval warm periods".

Note: Reviewer 1 comments start with RC1 while author responses start with AR (in blue).

RC1. General comments: The paper is interesting and reports worthwhile chronological and elemental data, interpreted in terms of recent intense ablation removing~600 years of ice from MPG (despite the glacier surviving warm times before then). Unfortunately, as presented, I am not convinced the data analysis supports the main conclusions and I cannot recommend the manuscript be published in its current form.

AR. We appreciate the interest on our manuscript and on the data presented. We provide our answers here to the three main points raised by Reviewer 1 hoping to solve his/her main concerns. Importantly, we approach the chronological issue in this new revised version following his/her advice regarding (1) the translation from lateral surface samples to depth, (2) explanations about the samples that we discarded from the chronology, (3) detailed interpretation of the "debris-rich" layers and (4) relation among the 100 studied samples and the three recovered ice cores. Ideas about the order and organization of the main text have been included in this new version of the manuscript and certainly have improved its readability.

RC1. First and most importantly, I believe the chronology needs to be addressed with greater structure and formal rigour. For example:

- Since this is so central to the paper's message, I find the translation from lateral surface samples to depth too difficult to follow in detail.

AR. We agree with this appreciation. Figure S2 showed a basic scheme of how we sampled the glacier ice and how we translate the surface samples to a depth profile. This figure was probably insufficient and not clear enough to understand our method. We provide an improved version of that figure (Figure 2 in the new manuscript). In that figure the bedding of the glacier ice is included with a downward slope of about 20 degrees. Although such a value has not been derived from local ice-thickness measurements, they are consistent with those derived from neighbouring GPR profiles further to the east (López-Moreno et al., 2019) . This figure will qualitatively help the readers to better understand the problem under study and how the sampling was done. A more complete information on this topic is included in the text of the revised version (section 3.1; lines 147-178). Please, see our responses to the following questions and the figures 1 and 2 of this letter to go further in depth into this topic.

RC1. It appears seven of 17 age samples were dismissed from the analysis; these need detailed comment on each.

AR. This information was already included in lines 309-327 of main text, but at a location within the paper that was not suitable. We greatly value the suggestions by the reviewer regarding restructuring of the manuscript contents, and have followed his/her suggestions. Of course providing the reasons to discard 7 out of 16 dates is one of the most important aspects of our discussion on chronology, so in the revised version we elaborate on the reasons to discard every sample (see text in lines 237-258). In addition, in Table 3 with all the 14C dates, we include a new column with comments regarding the quality of the dates.

Basically, we keep 8 samples from the first set of 9 bulk samples we dated. Those samples were the ones with more amount of organic material and we expected to get good 14C dates from them. The only one from that set we had to discard was MP46 that was out of order (younger than expected) and attributed to the presence of small pieces of plastic (too small to be removed) contaminating the sample. After obtaining the first 9 dates, we selected some other intervals where the dating information was insufficient but, unfortunately, the samples

were not so good in terms of quantity and quality of the organic material. Then, two samples dated by WIOC technique provided too large errors due to the low amount of carbon (still, we included one of them in the age model); three samples from pollen concentrates were too old (several millennia older than the others) likely due to associated problems to concentrate pollen (Kilian et al., 2002) and, finally, two samples from material contained in filters where the mixing among carbonate debris, dust and variable organic matter provided incongruent results. In summary, we constructed the age model from 8 samples from the first set of 9 bulk samples and 1 from WIOC. The other ones were not good enough to be considered in the chronology following the explained issues.

RC1. The interpretation of debris-rich englacial layers as periods of ablation (concentrating the debris) needs a far more rigorous argument based on physical analysis and exclusion of alternative possibilities. At present, the reader does not know whether these are isochronous, or whether they deform passively or cut-across primary layering/stratification. Could they be basally-derived? How are supply-rate variations excluded?

AR. Although alternative explanations cannot be completely excluded, we have firm reasons to believe that the sequence of debris-rich layers observed along the sampled profile correspond to the primary stratification of debris deposited at the surface of the glacier and are therefore isochronous layers (except for the cases in which the primary layers became merged due e.g. to intense melting episodes and/or low surface accumulation periods).

Note, first, that the distribution of layers is rather regular and extends laterally as shown in new Figure 2 in the manuscript, as would be expected for a stratification stemming from the original deposition at surface of snow and debris. Also note that the isochronal layers in a glacier emerge in the ablation area, but this is consistent with our case study, as our sampled profile corresponds entirely to the ablation zone. The reality is more complex, as our glacier has been shrinking and retreating since the end of the LIA, but the situation would be approximately as depicted in the attached figure (Figure 1).

[Figure 1 caption: In (1) we see the particle trajectories (blue) and the associated isochrones (red), emerging below the ELA; in (2) the trajectories have been removed to show only the

isochrones; (3) shows subsequent stages t1, t2, t3 of glacier shrinkage and retreat; (4) shows the current situation, similar to that illustrated in Figure 2 in the manuscript].

Additionally, we have prepared and run a dynamic flow model for a flowline as close as possible to our sampling profile using the available ground-penetrating radar data. Results are represented in Figure 2 where a collection of isochrones is shown in a similar way as they were represented schematically in Figure 2 of the manuscript. The limitation of this model is that it uses the current glacier geometry as if it was stationary during the whole modelled period. Still, it is interesting to see the approximate tilt of the ice bedding and the way in which the surface slope cuts the isochrone layers. If we carry out our sampling procedure along the slope represented here, we would sample the oldest material in the bottom part and the newest ice towards the top, as indicated by the area inside the black square. This illustrates perfectly the rationale of our sampling strategy.

[Figure 2 caption: Output of a dynamic flow model run in a closer area to the sampling transect where few georradar data were available. Colour lines are isochrones. The black square indicates a potential sampling transect along the slope cutting first the oldest ice beds and later the newest ones].

The reviewer questions whether the observed debris layers could be basally-derived. We believe that they are not. The main reason is that, in general, debris layers transverse to the glacier flow direction correspond to thrust faults developed near the glacier margin due to the compressional stress regime, most often associated to the transition, when we approach the glacier terminus, from warm-based to cold-based ice near the terminus. The warm-based ice slides over its bed, while the cold-based one does not, so strong compressional stresses develop, which cannot be accommodated by creep and thrusts develop. These thrust faults may reach the surface or terminate englacially (blind thrusts; see figure 3a below). Basal debris is incorporated into these thrusts, sometimes reaching the surface. When they do so, due to surface melting, they often produce large accumulations of debris (often in the form of pinacles or pinnacle ridges along the debris layer; see figs. 3a and 4 below); intense melting episodes can spread this debris over the glacier surface.

---

## Referee Report (RR1)

[referee-annotated manuscript omitted]

---

## Author Response (AR2)

Zaragoza, January 13th, 2021

Dear Dr. Farinotti,

I am submitting the revised version of our TC-2020-107 manuscript entitled "**The case of a southern European glacier which survived Roman and Medieval warm periods but is disappearing under recent warming**", co-authored by myself and colleagues, to be considered for publication in *The Cryosphere*.

In this new version we have incorporated the minor changes indicated by the two reviewers and provide a response to their main concerns that can be summarized as follows:

- Rev1 pointed to the lack of any value of 210Pb or 137Cs from other sites to be compared with ours. This is corrected and comparison is now much clear. Table 1 and 2 are also corrected following Rev1 comments.
- Rev1 was concern about our presentation of the principal result of this manuscript (the lack of ice from last 600 years) as a fact and not as an interpretation. We have carefully changed those sentences and, in general, our arguments in that regard have been moderated in this revised version.
- Rev2 was concern about the glacier bed structure in our Figs 2A and 5. This is changed since we don't have any information about the bedrock under the current ice and any idea of possible old ice trapped on "overdeepenings" is removed. Fig. 5 is also modified according Rev2 comments regarding the representation of different glacier stages, particularly we tried to reflect that the new ice pushes older ice masses from the accumulation to the ablation area where they may ablate.
- Rev2 noted that only 35 samples were analysed for trace elements, so their ID (position in the sequence) should be incorporated into Fig. 4 to make clear we don't observe the increase of Pb/Al or Hg associated to the Industrial Era. Since the incorporation of that information in Fig. 4 was difficult and made the figure less clear, we have included a new table with the sample IDs (Table 5).

We hope this new version was suitable for *The Cryosphere*, and hope that it will fulfil your expectations. I will be happy to answer any question you might have regarding this study.

Yours sincerely,

anenero

Ana Moreno Caballud

Avda. Montañana, 1005 50.059Zaragoza (España) TEL.:976-369-393 Answers to tc-2020-107 RC1 "The case of a southern European glacier disappearing under recent warming that survived Roman and Medieval warm periods".

Note: Reviewer 1 comments start with RC1 while author responses start with AR.

**RC1. General comments**

The revised manuscript reads more clearly and progresses in a more structured way than did the original version. The background and sampling strategy are also improved, and the rationale for sample exclusion/selection is included. Thus, the manuscript is greatly improved but I still have a couple of general concerns and requests, as well as a few specific ones (outlined below).

AR We greatly appreciate the comments made by Rev1 about the improvement of this version.

RC1. My first general comment is that the very low concentrations of Cs and Pb are taken as evidence that the ice being sampled is not recent – and that there is therefore a hiatus in the record and thus that modern ice, including that assumed to have formed during the LIA, has ablated. Given the sampling strategy I'm now ok with this argument – but it is missing one crucial element: that the Cs and Pb concentrations in recent ice elsewhere across southern Europe (or farther afield if none is available) are not summarized and presented for comparison. In other words, how do we know that the levels expected in recent ice are higher than those measured in MPG? I believe this step in the argument needs presenting formally, supported with data from other studies.

AR We agree about this issue. It is true that values from other locations were not included in the manuscript and the comparison with our values was then difficult. We have included now reference to other ice cores in European glaciers where 210Pb and 137Cs were measured to facilitate the comparison. In fact, all glacier surface samples across the European Alps present a similar 210Pb activity concentrations, on average  $86 \pm 16 \text{ mBq kg}^{-1}$  (Gäggeler et al., 2020), while ours are most of them below MDA and when is measurable is lower than 20 mBq kg-1. Regarding Cs values, MPG samples are all below MDA, while in other European glaciers are about 3 Bq kg-1 (Di Stefano et al., 2019).

RC1. Incidentally, Table 1 does not help here since it does not present the values measured in MPG (nor any from elsewhere in the literature) but instead has three columns: 'Sample', 'Mass of ice analysed' and 'MDA' (not defined in the caption – but is given in the text as 'minimum detection activity'). Thus, there doesn't seem to be a column for sample results (Table 2 – presenting Pb – does this (although MDA is again not defined)). I wonder whether Table 1 is missing a column; if it is not then the column labelled 'MDA' needs some explanation so the reader can follow the activity that was recorded.

AR Regarding this comment, we have included a new column in Table 1 to indicate the measured values of 137Cs, which were all below the MDA (definition is now in the caption).

RC1. Second, the manuscript includes sufficient uncertainty (for example, in terms of ice flow, the age-distance model, the origin of englacial debris, the explanation of some higher concentrations of Pb, the exclusion of certain samples etc.) that I believe the principal interpretation that no ice is present at the glacier that formed in the last ~600 years should in all cases be presented as interpretation – and not as fact. This is only a matter of appropriate wording, and this is already done in most cases – but not all (I note one or two below).

AR We agree with this appreciation and have changed our wording to reflect more clearly which sentences are just our interpretations and which ones reflect a fact.

RC1. Third, some interpretation is still presented in the Results section. In this case, these interpretations relate to my first point above and I believe the manuscript would be clearer if these comparisons (with concentrations in modern glacier ice) were removed from Results and dealt with specifically and separately in Interpretation/Discussion.

AR The change indicated by Rev1 has been carried out.

**RC1**. Specific comments

AR We have corrected all the typos and carried out all changes indicated by Rev1. Only our response to the reviewer comments that require more information from our side is included.

RC1. L49 – 54. I think this could be improved. How about: "The apparent absence of ice from the past ~600 years suggests that any ice accumulated during the Little Ice Age has since ablated. This interpretation is supported by measured concentrations of anthropogenic metals, including Zn, Se, Cd, Hg and Pb, which have concentrations well below those typical of industrial-age ice measured at other glaciers \*in the region\*. This study strengthens the general understanding that warming the past few decades has been exceptional for the past two millennia." (\*define as appropriate to the data presented\*).

AR We have included the sentences indicated by Rev1 since we agree about the data presentation and comparison with other glaciers. For example, comparing with the trace element data obtained from the glacier of Mt. Ortles, it is clear that the Enrichment Factors (EFs) reported for Zn (118), Ag (135), Bi (185), Sb (401) and Cd (514) are well above the crustal value, demonstrating the predominance of non-crustal depositions and suggesting an anthropogenic origin (Gabrieli et al., 2011). On the contrary, those elements are much lower than the crustal values in MPG (Table 4).

RC1. L159: (I don't follow the argument that the glacier being frozen to its base (incidentally, no robust evidence is presented to support this claim – only that the glacier is 'small') links to the clause 'to become of substantial age'. I think this sentence needs rewriting.

AR We include as an argument that there were evidences of no movement. Then, the glacier was expected to be frozen to bedrock.

RC1. L266: (47 mm here presumably refers to the diameter of the filter; however, it is the filter's pore size that is methodologically relevant.)

AR The reviewer is right and 47 mm refers to the diameter of the filter, a data which is probably not relevant. The filters we employ - Pallflex Tissuquartz™Filters - do not have a specific pore size since they are made of quartz fibers (see the webpage of the provider - www.pall.com/lab - with detailed description: "Binder-free pure quartz offers superior chemical purity. High flow rate and filtration efficiency. Uniquely designed for air monitoring in high temperatures and aggressive atmospheres"). In fact, this type is the purest one in terms of chemistry and able to retain basically everything.

RC1. Also, see comment above - some data are needed here to demonstrate the low activities.

AR A new column is included in Table 2. References to other sites are included.

RC1. L309 and L316 and 317 are all Interpretation and not Results

AR The low values obtained have to be included here as Results. Later, those values are interpreted in the Discussion section about age model.

RC1. L311-314: These are undetectable here – fine, but when these low levels are interpreted later they need to be compared with levels in recent ice; the argument that low levels means that the ice cannot be recent needs comparison data of recent ice with higher levels...

AR We agree and values of recent ice from other European glaciers are now included.

RC1. L352-353: (This is interpretation, not Results)

AR As indicated above, we present the results here and later use them to compare with other glaciers.

RC1. L370-371: (This is a bit awkward. How about: "Our age depth-model for MPG suggests the glacier is composed of ice that is up to ~2000 years old, and that the glacier's subsequent history has involved three main ..."). (Shouldn't this be 'four' main periods?)

AR We consider three periods since the last 600 years that could be period number four is not recorded in the ice, that is: we don't find any ice with that age.

**References**

Di Stefano, E., Clemenza, M., Baccolo, G., Delmonte, B. and Maggi, V.: 137Cs contamination in the Adamello glacier: Improving the analytical method, Journal of Environmental Radioactivity, 208–209, 106039, https://doi.org/10.1016/j.jenvrad.2019.106039, 2019.

Gabrieli, J., Carturan, L., Gabrielli, P., Kehrwald, N., Turetta, C., Cozzi, G., Spolaor, A., Dinale, R., Staffler, H., Seppi, R., dalla Fontana, G., Thompson, L. and Barbante, C.: Impact of Po Valley emissions on the highest glacier of the Eastern European Alps, Atmospheric Chemistry and Physics, 11(15), 8087–8102, https://doi.org/10.5194/acp-11-8087-2011, 2011.

Gäggeler, H. W., Tobler, L., Schwikowski, M. and Jenk, T. M.: Application of the radionuclide 210Pb in glaciology – an overview, Journal of Glaciology, 66(257), 447–456, https://doi.org/10.1017/jog.2020.19, 2020.

Answers to tc-2020-107 RC2 "The case of a southern European glacier disappearing under recent warming that survived Roman and Medieval warm periods".

Note: Reviewer 2 comments start with RC2 while author responses start with AR.

RC2 Comments by Wilfried Haeberli and Pascal Bohleber on "The case of a southern European glacier which survived Roman and Medieval warm periods but is now disappearing under recent warming" Revised paper submitted to The Cryosphere by Ana Moreno and 23 co-authors

**RC2 General**

The originally submitted paper is now available in a carefully revised version. The authors responded in a detailed and constructive way to the feedback and the recommendations from the side of the reviewers. The presentation of their comprehensive study about the age and composition of glacier remains at Monte Perdido (MPG) in the Pyrenees is now considerably improved and well worth publishing. A final polishing step, however, is still recommended. Besides the following general recommendations, an annotated file contains minor and rather technical comments.

AR. We appreciate very much these comments and the detailed work done in the annotated file by Rev2.

RC2. A new paper about ages of ice in an Alpine glacier was just published a few days ago:

Festi, D., Schwikowski, M., Maggi, V., Oeggl, K. and Jenk, T.M. (2020): Significant mass loss in the accumulation area of the Adamello glacier indicated by the chronology of a 46 m ice core. The Cryosphere Discussion. https://doi.org/10.5194/tc-2020-334 This study closely relates to the topic investigated at MPG. The authors may wish to read, discuss and cite this paper.

AR. Yes, we were aware of that paper and it is very good example of an ice core successfully dated by the 210Pb technique. Their figure 3 shows an excellent decaying profile, which is used to date the 46 meters of that ice sequence, reaching back to around 1944 AD. However, the accumulation rate and the melting rate are certainly different from our record in MPG. We have included this reference in the revised version of our manuscript as an example to compare their 210Pb concentrations with those obtained in our study.

RC2.Environmental conditions and age structure of the ice

The investigated site in a mountain permafrost environment (altitude, extreme shadow) is a perennial and quite probably cold "ice patch" or "glacieret" (most probably frozen to its bed), which developed from a considerably larger, warmer (probably polythermal) and faster moving (sliding) glacier which has been rapidly shrinking as a consequence of ongoing climate change. The internal age structure of the today remaining ice is the result of a highly complex transient development. Numerical modeling of such processes would require high-resolution spatiotemporal input parameters about changing ice geometries, mass balance or englacial temperatures, which are hardly available. In view of this difficulty and limitation, the presented results from near-surface probing together with the schematic concept of internal layering as illustrated in Figure 1 (best to be included in the paper?) of the author response file is good enough as a strongly simplified but quite reasonable first-order approximation. Such a simplified "first order approximation" should modestly be defined and treated as such. In this sense, the glacier bed in Figure 2A should be simplified and shown by a dashed line (for

uncertainty; perhaps even with question marks) if not precisely determined in situ by radar soundings. Especially the over-deepened part near the lower end of the ice patch must be dealt with in a physically sound way. If simply assumed or extrapolated from another site it would be better to eliminate it. If really and exactly documented in situ, the isochrones must be adjusted to reach the surface: ice in an overdeepening does not "flow into bedrock" as suggested now in Figure 2A but can move upslope at the ice base and reach the surface under the influence of local basal shear stresses as governed by surface slope and ice depth. This is more than a minor technical detail: With the presentation in Figure 2A the authors imply that passive and (much?) older ice is preserved in the overdeepening and overridden by younger ice along a mechanically questionable "shear zone". Even though this cannot be definitely excluded, there is no evidence visible for such conditions. Fundamental physics of glacier flow should also be more carefully considered in the interesting but extremely oversimplified Figure 5 (see comment in the annotated file).

AR. We agree with this comment and have modified the glacier bed in Figure 2A. Now it is represented by a dashed line, very simplified and without any "over-deepened parts". We think this new representation is adjusted to what we know and does not imply the presence of older ice since we don't have any evidence of it.

Figure 5 is more difficult to be modified since the schemes are smaller. Of course we agree about the fact that "layers of younger ice are not simply "put onto the surface" of already existing older ice but push older ice masses from the accumulation to the ablation area where they may ablate and disappear" and have tried to incorporate it to the figure. Additionally, we would like to note that we did not sample the upper glacier (too risky) and the colors indicating in Fig. 5 the age of the ice preserved were purely our "best guess" but not supported by any evidence. Those colors are now removed to avoid misinterpretation.

**RC2. Radiometric and glacio-chemical analyses**

The selection of the 14C data is now much better presented. Likewise, the presentation of the sample ID and the meaning of the depth increment used in the ice sampling has been revised and is now much clearer. However, one important point remains to be addressed: The authors use the sample ID to clearly reference the samples analyzed for 137Cs (Table 1), 210Pb (Table 2) and 14C (Table 3). The same information is still missing for the trace element and Hg datasets. In their response, the authors write "We do not have the depth information for previous Figure 3 (now figure 4) and including sample ID (from 0 to 100) appears now unnecessary." It would in fact be important to include the sample ID, for sake of completeness but even more so for the following reason: Both, the comparison with the Marboré Lake record as well as the argument for the absence of ice from the industrial period would be more convincing if the entire sample range had been measured. Since a subset of 35 samples (line 265 in the revised manuscript) has been selected for measurement, the location of the samples within the record matters. A particularly relevant question to still be answered by the authors is if the subset includes the potentially youngest portion of the record or not corresponding to the statement that "In particular, the lack of a Pb/Al peak characterizing the Industrial Period in the upper sequence of the MPG confirms the absence of the last two centuries in MPG ice record" (line 411). One could add here "as far as it becomes evident from the analyzed subset of samples in our record" – or something similar. The sample ID could be presented in form of a table (or adding to existing tables), or quite elegantly, as an additional x-axis to Figure 4.

AR. First, we would like to note that the subset of 35 samples was selected along the whole sequence, including of course samples from the upper part. To make all this issue more clearly indicated for the readers, we have added an additional table with the Pb/Al and Hg values for

MPG samples, indicating the sample ID. We tried to add the ID labels to Fig. 4 but it was difficult to make all the numbers easy to read and we have preferred the table format.

RC2. Some minor technical comments can be found in the annotated pdf.

AR. We have revised all those technical comments and acknowledge the efforts done by the reviewers to improve our manuscript.